# ToMoE: Converting Dense Large Language Models to Mixture-of-Experts through Dynamic Structural Pruning

**Shangqian Gao** *                                                                    *sg24bi@fsu.edu*
*Department of Computer Science, Florida State University*

**Hua Ting**                                                                            *thua@nd.edu*
*Department of Computer Science and Engineering, University of Notre Dame*

**Reza Shirkavand**                                                                     *rezashkv@cs.umd.edu*
*Department of Computer Science, University of Maryland, College Park*

**Chi-Heng Lin**                                                                        *chiheng.lin@samsung.com*
*Samsung Research America*

**Zheng Tang**                                                                          *zheng.tang@samsung.com*
*Samsung Research America*

**Zhengao Li**                                                                          *zl23i@fsu.edu*
*Department of Computer Science, Florida State University*

**Longge Yuan**                                                                         *ly23a@fsu.edu*
*Department of Computer Science, Florida State University*

**Fangyi Li**                                                                           *fangyili@seas.upenn.edu*
*School of Engineering and Applied Science, University of Pennsylvania*

**Zeyu Zhang**                                                                          *zeyzhan@amazon.com*
*Amazon AGI*

**Alireza Ganjdanesh**                                                                  *aliganj@umd.edu*
*Department of Computer Science, University of Maryland, College Park*

**Lou Qian**                                                                            *qian.lou@ucf.edu*
*Department of Computer Science, University of Central Florida*

**Xu Jie**                                                                              *xujie@ufl.edu*
*Department of Health Outcomes and Biomedical Informatics, University of Florida*

**Yen-Chang Hsu**                                                                       *yenchang.hsu@samsung.com*
*Samsung Research America*

**Reviewed on OpenReview:** *https://openreview.net/forum?id=RFHq46pjb6*

## Abstract

Large Language Models (LLMs) demonstrate remarkable capabilities but face deployment challenges due to their high computational demands. Traditional pruning methods reduce these costs by permanently removing parameters, which inevitably leads to performance degradation. To mitigate this issue, we propose ToMoE, a method that transforms dense LLMs into Mixture-of-Experts (MoE) models by uncovering experts inherently present

---

*Corresponding author.

within dense models, without requiring any weight updates. ToMoE leverages dynamic structural pruning to unify expert construction and router training in a single stage, achieving consistently strong performance. Remarkably, even without fine-tuning the model weights, ToMoE consistently outperforms state-of-the-art pruning and MoE techniques across Phi-2, LLaMA-2, LLaMA-3, and Qwen-2.5 models. The code for this paper is available at `https://github.com/gaosh/ToMoE`.

# 1 Introduction

Although LLMs demonstrate remarkable capacity to perform diverse tasks (Brown, 2020; Kenton & Toutanova, 2019; Raffel et al., 2020; OpenAI, 2022; Anthropic, 2023; Dong et al., 2022; Radford et al., 2019; Kaplan et al., 2020), their huge model size often limits their usability on devices with limited resources. As a result, considerable efforts (Ma et al., 2023; Ashkboos et al., 2024; Frantar et al., 2022) are focused on minimizing the computational and memory costs of these models. Structural pruning (Ma et al., 2023) has emerged as a promising solution to this challenge because, unlike unstructured pruning, it achieves compression without the need for specialized implementations. However, the problem with structural pruning methods is that they will substantially reduce the model capacity, resulting in an obvious performance gap compared to the dense model. The fine-tuning cost for even partially recovering this gap is tremendous.

To achieve a better trade-off between the number of parameters and performance, sparse Mixture of Experts (MoE) models (Shazeer et al., 2017; Lepikhin et al., 2021b) are designed to activate only a subset of the model's parameters, corresponding to the selected experts. Recently proposed MoE models, such as DeepseekMoE (Dai et al., 2024), demonstrated that they can match the performance of dense models with a similar total parameter count while using a small number of active parameters. Following this lead, transforming dense models into MoE models could offer a promising approach to bridging the performance gap left by structural pruning methods. Unlike prior efforts to construct MoE models from dense models (Zhang et al., 2022; Lee et al., 2024b; Zhu et al., 2024), our findings reveal that MoE inherently exists within dense models and can be uncovered without updating model weights (continue pretraining). Specifically, we show that these experts can be identified through dynamic structural pruning. These results represent a novel contribution that has not been demonstrated in previous studies.

The core idea of MoE models is conditional computation, where experts are dynamically selected based on input tokens. This concept aligns closely with dynamic pruning methods (Gao et al., 2019), which make pruning decisions given input features. Leveraging this connection, we propose to construct MoE models from dense models by using dynamic structural pruning. Specifically, for Multi-Head self-Attention (MHA) layers, we apply top-K routing and static pruning for compression, while for MLP layers, we transform them into MoE layers using top-1 expert routing. The routing mechanism learned for dynamic structural pruning can be directly applied to serve as the routing module for MoE layers. With differentiable discrete operations, the MoE conversion process can be formulated as a differentiable dynamic pruning problem. With this formulation, we can efficiently convert a dense model to an MoE model at a cost similar to or lower than regular structural pruning methods. The comparison between our method, static pruning, and the original LLM is shown in Fig. 1.

Built upon the above findings and techniques, we proposed ToMoE to effectively convert dense LLMs to MoE models with dynamic pruning. The contributions of this work can be summarized as follows:

- **Dense-to-MoE Conversion Through Dynamic Pruning:** We introduce a novel approach to convert dense models into MoE models through dynamic pruning. Specifically, we implement top-K routing and static pruning for MHA layers along the head dimension and top-1 routing for MLP layers across the learned experts. This formulation ensures sparse and efficient computation while retaining model capacity.
- **Joint Optimization for Routing and Experts:** The proposed method involves jointly optimizing routing modules and expert configurations by solving a regularized optimization problem. Our approach leverages differentiable operations to enable efficient and flexible MoE constructions.

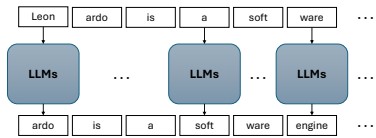
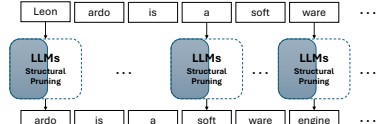
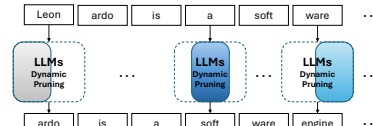

(a) Original LLMs    (b) LLMs with structural pruning.    (c) LLMs with dynamic pruning for MoE.

Figure 1: **(a):** The original LLM uses all parameters to process all the input text. **(b):** The static pruning for LLMs permanently removes model parameters, and the rest of the parameters are used to process all the input text. **Our approach (c):** LLMs with dynamic pruning use different sub-networks (illustrated by different colors) to process different tokens. We incorporate MoE to achieve a fixed expected budget in inference.

- **Consistent Performance Improvements:** Our method consistently outperforms state-of-the-art structural pruning and MoE construction techniques on various tasks while training only the router, without fine-tuning the model weights. This performance improvement is demonstrated across widely used public models such as Phi-2, LLaMA-2, LLaMA-3, and Qwen-2.5.
- **Detailed Analysis:** We extensively analyze the resulting model from ToMoE across multiple perspectives, including parameter allocation, router behavior, and the ablation of different design components. We hope these analyses provide valuable insights and guidance for future research in this area.

## 2 Related Works

**Pruning:** Structural pruning (Li et al., 2017; Kurtic et al., 2022; Ma et al., 2023) is an attractive technique for real-world deployments since it removes redundant parameters to reduce model size without requiring specialized implementations. Structural pruning methods fall into two main categories: static pruning (Anwar et al., 2017; Molchanov et al., 2019; Fang et al., 2023) and dynamic pruning (Gao et al., 2019; Chen et al., 2020; Anagnostidis et al., 2023; Dong et al., 2024a). Static pruning removes parameters based on input-agnostic importance metrics. For example, LLM-Pruner (Ma et al., 2023) eliminates non-essential coupled structures using gradient-based criteria. The problem with structural pruning is that it often creates a noticeable performance gap relative to dense models (Ma et al., 2023; Ashkboos et al., 2024). In contrast, dynamic pruning removes weights based on input-dependent metrics. Early attempts for dynamic pruning (Gao et al., 2019; Chen et al., 2020) focus on Convolutional Neural Networks, where channels are selectively activated for input samples. Recent works, such as D-LLM (Wang et al., 2024), incorporate the concept of conditional computation into LLMs by selectively skipping layers based on input tokens. The problem with dynamic pruning methods is that they do not have a fixed budget given different inputs, which creates problems when serving LLMs in a mini-batch setting or in the prefilling stage. Our method, on the other hand, converts the dense LLM to a sparse MoE model with a fixed per-token budget.

Another line of research applies contextual sparsity for LLMs (Liu et al., 2023; Zheng et al., 2024; Lee et al., 2024a), where neurons are selectively activated given certain conditions. Although there are some promising results, they are generally more difficult to achieve better inference efficiency due to their irregular sparsity patterns. In contrast, MoE models have more comprehensive support from the system side, making them a popular choice for scaling up the model. Thus, our method mainly focuses on converting dense models to MoE models.

**MoE:** Sparse Mixture-of-Experts (MoE) models improve upon pure structural pruning by maintaining or even enhancing model capacity without a proportional increase in computational cost. For instance, Sparsely-Gated MoE (Shazeer et al., 2017) employs a trainable gating network to select a small subset of experts for each input, enabling the model to scale to thousands of experts efficiently (Lepikhin et al., 2021b). More recent methods like DeepSeekMoE (Dai et al., 2024) further address expert specialization, matching dense-model performance with a similar number of activated parameters. Previous methods constructing MoE from the dense model (Zhang et al., 2022; Lee et al., 2024b; Zhu et al., 2024) separate the expert construction and

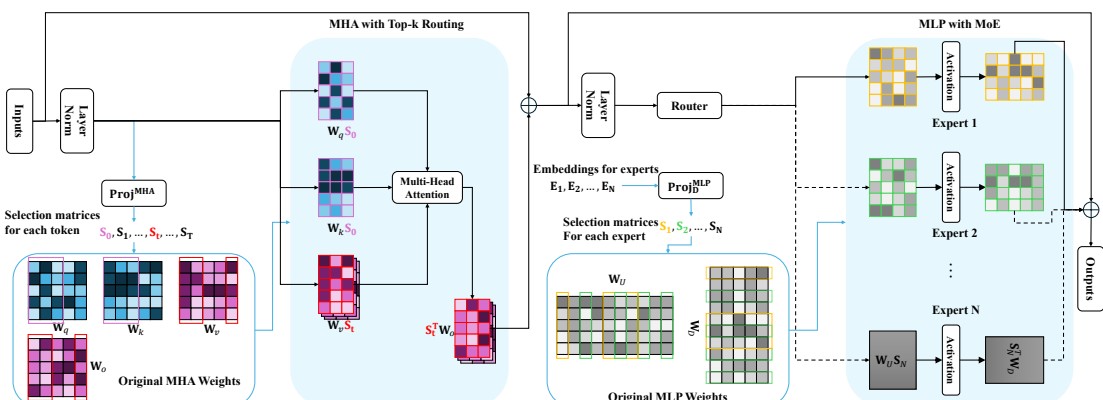

Figure 2: **ToMoE** uses top-1 routing for MLP layers, and static and dynamic pruning along the head dimension for MHA layers.

router training into two distinct stages, often leading to sub-optimal performance. In contrast, our method integrates expert construction directly into the pruning process, treating it as a unified step with router learning and thereby largely improving the performance without fine-tuning.

## 3 ToMoE

Most recent LLMs, like GPT (Radford et al., 2018), LLaMA (Touvron et al., 2023a), etc., adapt decoder-only architectures and thus our method focuses on decoder-only architectures. A typical decoder block consists of Multi-Head Attention (MHA) and Multi-Layer Perceptron (MLP) layers. For clarity, we denote the sequence length by $T$, the hidden dimension by $d$, the MLP intermediate dimension by $d_{\mathrm{mid}}$, and the number of attention heads by $H$.

To reduce the computational costs of the decoder-only architecture, we propose to convert the original model into MoE models. For MHA layers, we utilize top-K routing and static pruning along the head dimension $\frac{d}{H}$. Top-K routing and static pruning for MHA layers ensure that, during prefilling or model serving, all tokens maintain the same head dimension, enabling parallel processing. For MLP layers, our approach transforms them into MoE layers along the MLP middle dimension $d_{\mathrm{mid}}$ and employs top-1 routing. A key distinction between our method and previous dynamic pruning approaches is that the converted model maintains consistent computational costs for all inputs. This property could be crucial for efficient processing.

### 3.1 Expert Embeddings

Inspired by the recent success of using hypernetworks (Ha et al., 2016; Ganjdanesh et al., 2024; Gao et al., 2024) to generate pruning decisions, we adopt a hypernetwork to generate expert embeddings:

$$\mathbf{E}_{\mathrm{all}} = \mathrm{HN}(z), \tag{1}$$

where $z$ is the input to the hypernetwork drawn from a random distribution, and $\mathbf{E}_{\mathrm{all}} = [\mathbf{E}_1, \cdots, \mathbf{E}_l, \cdots, \mathbf{E}_L]$ contains embeddings for all layers and $\mathbf{E}_l \in \mathbb{R}^{N \times d_e}$, where $N$ is the number of experts and $d_e$ is the expert embedding dimension. Each embedding $\mathbf{E}_{l,i}$ will then be used to generate the configurations of experts. The purpose of having the hypernetwork to generate $\mathbf{E}_{\mathrm{all}}$ is to introduce inter-layer dependencies across different layers and operations. This design has been shown to accelerate the learning process in practice (Gao et al., 2024). More details are given in the Appendix A.

### 3.2 Expert Construction

In this section, we will talk about how to construct experts from MLP layers. In a decoder layer, the formulation of MLP is: $f_{\mathrm{MLP}}(\mathbf{X}) = \sigma(\mathbf{X}\mathbf{W}_G) \odot (\mathbf{X}\mathbf{W}_U)\mathbf{W}_D$, where matrices $\mathbf{W}_U \in \mathbb{R}^{d \times d_{\mathrm{mid}}}$, $\mathbf{W}_G \in \mathbb{R}^{d \times d_{\mathrm{mid}}}$

and $\mathbf{W}_D \in \mathbb{R}^{d_{\mathrm{mid}} \times d}$ denote up, gated, and down projection matrices. In addition, $\sigma$ denotes nonlinear activation functions and $\odot$ denotes the Hadamard product (element-wise product).

Assume the target is to use $N$ experts, under the setting of structural pruning, each expert can be represented by:

$$f_{\mathrm{MLP}}^i(\mathbf{X}_t) = \sigma(\mathbf{X}_t \mathbf{W}_G \mathbf{S}_i) \odot (\mathbf{X}_t \mathbf{W}_U \mathbf{S}_i) \mathbf{S}_i^\top \mathbf{W}_D, \tag{2}$$

where $i = 1, \cdots, N$, and $\mathbf{S}_i = \mathrm{Diag}(\mathbf{s}_i)$ ($\mathbf{s}_i \in \mathbb{R}^{d_{\mathrm{mid}}}$, $\mathbf{S}_i \in \mathbb{R}^{d_{\mathrm{mid}} \times d_{\mathrm{mid}}}$), is a binary diagonal matrix selecting a subset of weight vectors for the $i$th expert. $\mathbf{X}_t$ is the $t$th token, which is assumed to be routed to the $i$th expert. Once each expert is formulated, its configuration is learned as follows:

$$\mathbf{s} = \mathrm{ST\text{-}GSig}(\mathrm{Proj}_D^{\mathrm{MLP}}(\mathbf{GE})), \quad \mathbf{G} = \mathrm{ST\text{-}GSmax}(\mathrm{Router}(\mathbf{X})), \tag{3}$$

where $\mathbf{G} \in \mathbb{R}^{T \times N}$ is the output of the router module, $\mathbf{E}$ ($l$ is omitted for clarity) is the expert embeddings, $\mathrm{Proj}_D^{\mathrm{MLP}} : \mathbb{R}^{d_e} \to \mathbb{R}^{d_{\mathrm{mid}}}$ is a projection module to project the latent embedding to the MLP middle dimension, $\mathrm{Router}(\cdot) : \mathbb{R}^{d_e} \to \mathbb{R}^N$ is the router module that maps the inputs to an $N$-dimensional routing score vector for expert selection, and ST-GSig and ST-GSmax are Straight-Through Gumbel-Sigmoid and Gumbel-Softmax functions respectively (Jang et al., 2016). Under this setting, $\mathbf{s}_i$ will contain retained positions (represented by 1) for the $i$th expert, and $\mathbf{G}$ contains one-hot routing decisions for tokens in $\mathbf{X}$.

### 3.3 MHA top-K Routing

An MHA layer can be represented as $f_{\mathrm{MHA}}(\mathbf{X}) = \sum_{i=1}^H \sigma_s \left( e(\mathbf{X}\mathbf{W}_{Q,i}) e^\top(\mathbf{X}\mathbf{W}_{K,i}) \right) \mathbf{X}\mathbf{W}_{V,i} \mathbf{W}_{O,i}$, where $\mathbf{W}_{Q,i}, \mathbf{W}_{K,i}, \mathbf{W}_{V,i} \in \mathbb{R}^{d \times \frac{d}{H}}$, $\mathbf{W}_{O,i} \in \mathbb{R}^{\frac{d}{H} \times d}$ are the query, key, value, and output matrices for each attention head, and $\mathbf{X} \in \mathbb{R}^{T \times d}$ is the input hidden states. $e$ and $\sigma_s$ denote positional embedding and the softmax function.

For MHA layers, we perform two kinds of pruning: dynamic top-K pruning and static pruning, both along the head dimension. Justifications regarding the design choice are provided in the Appendix F. Like MLP layers, we also insert selection matrices:

$$f_{\mathrm{MHA}}(\mathbf{X}_t) = \sum_{i=1}^H \left[ \sigma_s \left( e(\mathbf{X}_t \mathbf{W}_{Q,i}) \mathbf{S}_0 \mathbf{S}_0^\top e^\top(\mathbf{X}_t \mathbf{W}_{K,i}) \right) \mathbf{X}_t \mathbf{W}_{V,i} \mathbf{S}_t \right] \mathbf{S}_t^\top \mathbf{W}_{O,i}, \tag{4}$$

where $\mathbf{S}_0, \mathbf{S}_t \in \mathbb{R}^{\frac{d}{H} \times \frac{d}{H}}$ are selection matrices. $\mathbf{S}_0$ is the shared selection matrix for static pruning of query and key matrices, while $\mathbf{S}_t$ is the token-specific selection matrix for the value and output matrices of the $t$-th token. We apply the same selection matrix across all heads, ensuring that all heads have the same head dimensions at inference time. To generate the selection matrix, we calculate its diagonal vector $\mathbf{s}_t$ as:

$$\mathbf{s}_t = \mathrm{ST\text{-}GSig}(\mathrm{Proj}_D^{\mathrm{MHA}}(\mathrm{Proj}_E^{\mathrm{MHA}}(\mathbf{X}_t) + \frac{1}{N}\mathbf{1}^\top \mathbf{E})), \tag{5}$$

where $\mathbf{1} \in \mathbb{R}^N$ is an all-one vector, $\frac{1}{N}\mathbf{1}^\top\mathbf{E}$ represents the average expert embedding of size $d_e$, $\mathrm{Proj}_D^{\mathrm{MHA}} : \mathbb{R}^{d_e} \to \mathbb{R}^{\frac{d}{H}}$ is a projection module to map the latent embedding to the head dimension and $\mathrm{Proj}_E^{\mathrm{MHA}} : \mathbb{R}^d \to \mathbb{R}^{d_e}$ is also a projection module to project input tokens to the space of expert embeddings, and ST-GSig is defined in Sec. 3.2. When $t = 0$, we initialize $\mathbf{X} = 0$, and set $\mathbf{s}_0 = \mathrm{ST\text{-}GSig}(\mathrm{Proj}_D^{\mathrm{MHA}}(\frac{1}{N}\mathbf{1}^\top\mathbf{E}))$, since it is input independent.

During training, the number of ones in $\mathbf{s}$ can vary freely. After training is complete, we compute $K = \mathrm{round}(\frac{1}{T}\sum_{t=1}^T \sum_{i=1}^{\frac{d}{H}} \mathbf{s}_{t,i})$ for a subset of tokens and use it for top-K routing during inference. Note that the $k$ in top-K for $\mathbf{s}_0$ and $\mathbf{s}_t$ ($t \geq 1$) can be different, allowing for larger flexibility. Also, note that $\mathbf{s}_0$ must follow specific structural constraints to be compatible with the position embedding $e(\cdot)$, and more details can be found in Appendix A.2.

### 3.4 Regularizations for MoE Constructions

In Sec. 3.2 and Sec. 3.3, we briefly introduced the design space for constructing MoE models using dynamic structural pruning. In this subsection, we will introduce regularizations customized to the characteristics of MoE models.

**Union of Experts Regularization.** An ideal sparse MoE model converted from a dense model should maximize parameter utilization, which means that the total number of parameters in the MoE model should closely approximate the dense model. To add this regularization to our learning process, we push the union of experts to be closer to the original model. More specifically, we use MHA layers as an example:

$$\mathbf{u} = \bigcup_{i=1}^{T} \mathbf{s}_i = 1 - \prod_{i=1}^{T}(1 - \mathbf{s}_i), \tag{6}$$

where $\bigcup$ is the union operator, and $\mathbf{u}$ is the union of all kept positions for each token. For MLP layers, it can be calculated similarly. We then push $\frac{\sum \mathbf{u}}{|\mathbf{u}|}$ ($|\mathbf{u}|$ represents the size of $\mathbf{u}$) to 1:

$$\mathcal{R}_{\mathrm{U}} = \frac{1}{L} \sum_{l=1}^{L} f_{\mathrm{reg}}\left(\frac{\sum \mathbf{u}_l}{|\mathbf{u}_l|}, 1\right), \tag{7}$$

where $f_{\mathrm{reg}}(\cdot, \cdot)$ can be any regression loss functions, and we will choose $f_{\mathrm{reg}}$ later.

**Parameter Regularization.** For a sparse MoE model, we also need to control the number of active parameters given the provided budget. To achieve this goal, we can directly penalize the maximum width across different experts. We choose the maximum width over experts instead of the mean, median, or other alternatives because the maximum provides precise control over the upper bound of the number of active parameters.

Denote the width of a layer as $d_l^*$, where $* \in \{\mathrm{MLP}, \mathrm{MHA}\}$. For MLP layers, it can be calculated by $d_l^{\mathrm{MLP}} = \max(\mathbf{s1}_{d_{\mathrm{mid}}})$, where $\mathbf{1}_{d_{\mathrm{mid}}} \in \mathbb{R}^{d_{\mathrm{mid}}}$ is an all-one vector of size $d_{\mathrm{mid}}$. $\mathbf{s1}_{\mathrm{mid}}$ produces the width of all experts, and $d_l^{\mathrm{MLP}}$ represents the maximum width across all experts. The width of MHA layers can be calculated similarly. Based on $d_l^*$, we can calculate the number of active parameters in the model $\mathrm{T}(\mathbf{d}_{\mathrm{MoE}})$, where $\mathbf{d}_{\mathrm{MoE}} = [d_1^*, \cdots, d_L^*]$. To push the number of active parameters to a predefined rate $p$, the following objective is applied:

$$\mathcal{R}_{\mathrm{P}} = f_{\mathrm{reg}}(\mathrm{T}(\mathbf{d}_{\mathrm{MoE}}), p\mathrm{T}_{\mathrm{total}}), \tag{8}$$

where $\mathrm{T}_{\mathrm{total}}$ is the total number of parameters, and $p \in (0, 1]$ represents the ratios of the active parameters. For $f_{\mathrm{reg}}$ in Eq. 8 and Eq. 7, the following function $f_{\mathrm{reg}}$ is used:

$$f_{\mathrm{reg}}(x, y) = \log(\max(x, y) / \min(x, y)).$$

**Load Balancing Regularization.** When determining the configurations of experts, we also apply the load balancing regularization to encourage a balanced load across experts (Lepikhin et al., 2021a; Fedus et al., 2022). The load balancing loss from the Switch Transformer (Fedus et al., 2022) is adopted:

$$\mathcal{R}_{\mathrm{L}} = N \sum_{i=1}^{N} F_i P_i, \tag{9}$$

where $F_i = \frac{1}{T} \sum_{t=1}^{T} \mathbb{1}(\mathbf{G}_{t,i} = 1)$. The indicator function $\mathbb{1}(\cdot)$ returns 1 if the condition is true and 0 otherwise. $F_i$ represents the fraction of tokens assigned to the $i$-th expert. $P_i = \frac{1}{T} \sum \mathrm{GSmax}(G_{t,i})$, where $G = \mathrm{Router}(\mathbf{X})$ is the router output **before ST-GSmax**. $P_i$ is the fraction of the router probability allocated for the $i$-th expert. $\mathcal{R}_{\mathrm{L}}$ will encourages uniform routing across different experts as shown in (Fedus et al., 2022). The combination of Eq. 8 and Eq. 9 creates an interesting phenomenon where they encourage uniform allocation of width among experts.

### 3.5 Learning to Construct MoEs

Based on the aforementioned techniques, MoEs can be constructed from the dense LLMs by training router parameters, projection parameters, and hypernetwork parameters, while keeping all original model parameters frozen. This approach enables the rapid construction of an effective MoE model with a resource budget

Table 1: Perplexity comparisons of structured pruning methods and ToMoE for LLaMA-2 7B and 13B models on WikiText-2.

| Method | LLaMA-2 7B (ppl: 5.12 ↓) | | | LLaMA-2 13B (ppl: 4.57 ↓) | | |
|---|---|---|---|---|---|---|
| | 70% | 60% | 50% | 70% | 60% | 50% |
| LLM-Pruner (Ma et al., 2023) | 13.56 | 17.90 | 31.05 | 12.19 | 19.56 | 32.20 |
| LLM Surgeon (van der Ouderaa et al., 2024) | 7.83 | 10.39 | 15.38 | 6.21 | 7.25 | 9.43 |
| ShortGPT (Men et al., 2024) | 33.21 | 71.04 | 268.11 | 30.48 | 48.83 | 187.23 |
| SLEB (Song et al., 2024) | 11.23 | 29.10 | 103.38 | 8.24 | 11.76 | 27.67 |
| SliceGPT (Ashkboos et al., 2024) | 10.47 | 15.19 | 24.82 | 8.68 | 12.56 | 20.57 |
| ModeGPT (Lin et al., 2024) | 7.51 | 8.41 | 11.88 | 6.10 | 6.95 | 8.95 |
| DISP-LLM (Gao et al., 2024) | 6.85 | 8.11 | 9.84 | 5.77 | 6.59 | 7.11 |
| **ToMoE (ours)** | **6.41** | **7.17** | **8.36** | **5.54** | **6.06** | **6.78** |

comparable to that of structural pruning. The overall framework of our method is shown in Fig. 2, and the corresponding training objective function can be formulated as:

$$\min_{\Theta} \ \mathcal{L}(f'(x; \mathbf{E}_{\text{all}}), f(x)) + \alpha \mathcal{R}_{\mathbf{P}} + \beta \mathcal{R}_{\mathbf{U}} + \gamma \mathcal{R}_{\mathbf{L}}, \qquad (10)$$

where $\Theta = [\Theta_{\text{HN}}, \Theta_{\text{Router}}, \Theta_{\text{Proj-MHA}}, \Theta_{\text{Proj-MLP}}]$, $\Theta_{\text{HN}}$ is trainable parameters for the hypernetwork in Eq. 1, $\Theta_{\text{Router}}$ and $\Theta_{\text{Proj-MLP}}$ are trainable parameters for the router and the project module in Eq. 3, $\Theta_{\text{Proj-MHA}}$ is the trainable parameters of the projection modules given in Eq. 5, $\mathcal{R}_{\mathbf{P}}$, $\mathcal{R}_{\mathbf{U}}$, and $\mathcal{R}_{\mathbf{L}}$ are regularization terms defined in Sec. 3.4. And $\alpha$, $\beta$, and $\gamma$ are hyperparameters to control the strength of these regularization terms. Here, $f$ represents the

Table 2: Comparisons with semi-structured pruning on LLaMA-2.

| Method | Structure | 50% (7B) | 50% (13B) |
|---|---|---|---|
| SparseGPT (2:4) | Semi-structured | 10.17 | 8.32 |
| Wanda (2:4) | Semi-structured | 11.02 | 8.27 |
| Pruner-Zero (2:4) | Semi-structured | 10.52 | 7.41 |
| ToMoE (ours) | Structured | **8.36** | **6.78** |

original dense model, and $f'$ is the model equipped with our designed modules for MoE construction. Under this setting, we use $\mathcal{L}(\cdot, \cdot)$ to calculate the KL divergence between the logits of $f$ and $f'$, which is used as the guidance to preserve the capacity of the dense model (Hinton et al., 2015). We also found that using the KL divergence alone can lead to the best performance, and this observation complies with the experimental setup in (Muralidharan et al., 2024). Also, note that we perform **in-place** knowledge distillation since the original model weights are frozen. Thus, the knowledge distillation process does not introduce overheads in terms of GPU memory.

After learning how to construct the MoE, we convert the MLP layer to $N$ experts with shared weights. After pruning the MLP layer, we save $\frac{1}{N}\mathbf{1}^{\top}\mathbf{E}$ for MHA layers as the bias of the $\text{Proj}_E^{\text{MHA}}$, and we drop $\text{Proj}_D^{\text{MLP}}$ and convert Eq. 5 into a Top-K routing function as well as use $\mathbf{s}_0$ for pruning $\mathbf{W}_Q$ and $\mathbf{W}_K$. Our construction also enables converting the MoE back into a pseudo-MoE model. The MoE model and the pseudo-MoE model are equivalent, and more details can be found in Appendix B.3.

## 4 Experiments

### 4.1 Settings

**Models.** Our ToMoE method is evaluated using several LLMs with decoder blocks. Specifically, we choose the following models: LLaMA-2 (Touvron et al., 2023b): LLaMA-2 7B and LLaMA-2 13B; LLaMA-3 8B (Dubey et al., 2024); Phi-2 (Javaheripi et al., 2023); Qwen-2.5 (Yang et al., 2024): Qwen-2.5 7B and Qwen-2.5 14B. Results for LLaMA-2 13B and Qwen-2.5 14B are presented in the Appendix D.

**Implementations.** ToMoE is implemented by Pytorch (Paszke et al., 2019) and Hugging Face transformer library (Wolf et al., 2020). The model weights are frozen when training the modules with learnable parameters $\Theta$ in Obj. 10. We use the AdamW (Loshchilov & Hutter, 2019) optimizer to optimize $\Theta$, which is trained for 10,000 iterations for all models. For all experiments, we set $\alpha = 16$, $\beta = 2.0$, and $\gamma = 1.0$, where $\alpha$, $\beta$, and $\gamma$ are defined in Obj. 10. Without specific descriptions, the number of experts for ToMoE is 8 across all

Table 3: Zero-shot task performance of compressed LLaMA-2 7B, LLaMA-3 8B, Qwen-2.5 7B.

| Model | Active Parameters | Method | ARC-e | ARC-c | PIQA | WinoG. | HellaS. | Average |
|---|---|---|---|---|---|---|---|---|
| | | | acc-norm | acc-norm | acc-norm | acc | acc-norm | |
| LLaMA-2 7B | 100% | Dense | 74.58 | 46.25 | 79.11 | 69.06 | 75.99 | 69.00 |
| | 60% | ShortGPT (Men et al., 2024) | 41.16 | 29.94 | 60.12 | 60.46 | 43.67 | 47.07 |
| | | SliceGPT (Ashkboos et al., 2024) | 36.49 | 24.57 | 54.90 | 53.43 | 34.80 | 40.84 |
| | | LLM Surgeon (van der Ouderaa et al., 2024) | 52.31 | 30.29 | 69.26 | 54.38 | 48.04 | 50.86 |
| | | ModeGPT (Lin et al., 2024) | 49.45 | 30.03 | 64.96 | 61.96 | 53.01 | 51.88 |
| | | ModeGPT-Alpaca (Lin et al., 2024) | 59.76 | 34.73 | 70.35 | **64.40** | 58.63 | 57.58 |
| | | **ToMoE (Ours)** | **63.64** | **38.74** | **72.85** | 62.51 | **65.84** | **60.72** |
| LLaMA-3 8B | 100% | Dense | 77.69 | 53.58 | 80.63 | 72.69 | 79.16 | 72.75 |
| | 75% | ShortGPT-Alpaca (Men et al., 2024) | 38.13 | 31.40 | 60.94 | 54.22 | 31.52 | 43.24 |
| | | SliceGPT-Alpaca (Ashkboos et al., 2024) | 44.44 | 29.27 | 57.56 | 58.48 | 41.08 | 46.17 |
| | | ModeGPT-Alpaca (Lin et al., 2024) | 67.05 | 41.13 | 75.52 | **69.61** | 66.49 | 63.96 |
| | 70% | **ToMoE (Ours)** | **71.55** | **44.71** | **76.28** | 68.98 | **71.84** | **66.67** |
| | 60% | **ToMoE (Ours)** | 65.87 | 41.64 | 73.61 | 63.30 | 66.42 | 62.17 |
| Qwen-2.5 7B | 100% | Dense | 79.42 | 50.17 | 79.54 | 71.35 | 78.36 | 71.77 |
| | 50% | DISP-LLM Alpaca (Gao et al., 2024) | 55.35 | 34.22 | 70.29 | 53.59 | 55.00 | 53.69 |
| | | **ToMoE N=8 (Ours)** | 61.83 | 36.77 | 71.82 | 57.70 | 59.55 | 57.53 |
| | | **ToMoE N=16 (Ours)** | **64.81** | **41.72** | **73.45** | 58.48 | 61.06 | **59.90** |
| | | **ToMoE N=24 (Ours)** | 64.69 | 39.85 | 72.96 | **58.56** | **63.21** | 59.85 |
| | 40% | DISP-LLM Alpaca (Gao et al., 2024) | 52.65 | 27.99 | 65.94 | 53.43 | 44.11 | 48.82 |
| | | **ToMoE N=8 (Ours)** | 53.80 | 32.59 | 69.59 | 55.49 | 52.98 | 52.89 |
| | | **ToMoE N=16 (Ours)** | 53.87 | **33.70** | **71.98** | 54.38 | **55.65** | 53.92 |
| | | **ToMoE N=24 (Ours)** | **55.43** | 32.85 | 69.86 | **57.85** | 54.62 | **54.12** |

(a) Loss $\mathcal{L}$  (b) Loss $\mathcal{R}_{\mathbf{P}}$  (c) Loss $\mathcal{R}_{\mathbf{U}}$  (d) Loss $\mathcal{R}_{\mathbf{L}}$

Figure 3: The training dynamics give different ratios $p$ of active parameters on the Qwen-2.5 7B model.

settings. Depending on the size of the base model, 1 to 4 NVIDIA A100 GPUs are used to train $\Theta$. More implementation details can be found in Appendix C.

**Datasets.** Two training settings are provided for all modules with learnable parameters $\Theta$: (1) using WikiText Merity et al. (2016), and (2) using a mixed dataset comprising WikiText, Alpaca Taori et al. (2023), and Code-Alpaca Chaudhary (2023) (mixing ratio: 1:1:1). Based on our observation, ToMoE benefits from a diverse mixture of datasets to effectively construct experts. Following previous methods (Ashkboos et al., 2024; Gao et al., 2024), our method and other methods are evaluated on five well-known zero-shot tasks: PIQA (Bisk et al., 2020); WinoGrande (Sakaguchi et al., 2021); HellaSwag (Zellers et al., 2019); ARC-e and ARC-c (Clark et al., 2018). We further evaluate our method on the following tasks and configurations to ensure consistency with comparison baselines: 32-shot BoolQ (Clark et al., 2019), SciQ (Johannes Welbl, 2017), 5-shot WinoGrande, 25-shot ARC-c, 10-shot HellaSwag, TruthfulQA Lin et al. (2022), and 5-shot MMLU (Hendrycks et al., 2021). We use llm-eval-harness (Gao et al., 2021) to evaluate the compressed models.

**Baselines.** ToMoE is compared to baselines from structural pruning methods (Ma et al., 2023; Ashkboos et al., 2024; Men et al., 2024; Song et al., 2024; van der Ouderaa et al., 2024; Lin et al., 2024; Gao et al., 2024), semi-structural pruning methods (Frantar & Alistarh, 2023; Sun et al., 2024; Dong et al., 2024b) and MoE construction methods (Zhu et al., 2024; Lee et al., 2024b; Qu et al., 2024; Pei et al., 2025).

Table 4: Compassion against MoE construction methods.

| Model | Active Parameters | Method | ARC-e | ARC-c | PIQA | WinoG. | HellaS. | Average |
|---|---|---|---|---|---|---|---|---|
| | | | acc-norm | acc-norm | acc-norm | acc | acc-norm | |
| LLaMA-2 7B | 100% | Dense | 74.58 | 46.25 | 79.11 | 69.06 | 75.99 | 69.00 |
| | 50% | LLaMA-MoE E16A4 (Zhu et al., 2024) | 27.02 | 29.53 | 49.13 | 49.49 | 26.19 | 36.27 |
| | | + fine-tuning | 35.52 | 26.88 | 49.13 | 55.71 | 34.07 | 40.55 |
| | | LLaMA-MoE E8A2 (Zhu et al., 2024) | 25.76 | 27.22 | 50.92 | 47.51 | 26.02 | 35.49 |
| | | + fine-tuning | 37.71 | 27.56 | 58.11 | 51.78 | 36.37 | 42.31 |
| | | **ToMoE (Ours)** | **56.65** | **33.87** | **71.00** | **58.56** | **60.26** | **56.07** |
| Phi-2 | 100% | Dense | 78.24 | 54.01 | 79.11 | 75.61 | 73.86 | 72.17 |
| | 78% | G-MoEfication (Lee et al., 2024b) | 66.46 | 38.91 | 70.29 | 65.75 | 57.90 | 59.86 |
| | 70% | G-MoEfication (Lee et al., 2024b) | 57.95 | 35.24 | 64.80 | 60.30 | 49.07 | 53.47 |
| | | **ToMoE (Ours)** | **70.79** | **43.86** | **77.00** | **66.38** | **62.68** | **64.16** |

Table 5: Comparison against MoE construction methods on LLaMA-2 7B.

| Active Parameters | Method | # Tokens | BoolQ (32) | SciQ | PIQA | WinoG. (5) | ARC-C (25) | HellaS. (10) | Average |
|---|---|---|---|---|---|---|---|---|---|
| | | | acc | acc | acc | acc | acc-norm | acc-norm | |
| 100% | LLaMA-2 7B | 2T | 82.04 | 90.80 | 78.78 | 73.95 | 53.15 | 78.55 | 76.21 |
| 50% | LLaMA-MoE E8A2 (Zhu et al., 2024) | 1.2B | 37.83 | 20.00 | 49.73 | 50.12 | 25.79 | 26.18 | 34.27 |
| | LLaMA-MoE-v2 E8A2 (Qu et al., 2024) | 1.2B | 51.25 | 67.00 | 56.64 | 52.88 | 25.68 | 35.10 | 48.59 |
| | CMoE E8S1A1 (Pei et al., 2025) | **0** | 46.09 | 65.30 | 52.77 | 48.70 | 23.80 | 30.12 | 44.13 |
| | + fine-tuning | 1.2B | 55.04 | 77.50 | 57.12 | 54.06 | 27.56 | 38.79 | 51.68 |
| | **ToMoE (Ours)** | 0.02B | **62.42** | **81.70** | **71.00** | **60.41** | **34.39** | **60.54** | **61.39** |

Table 6: Comparison against LLaMA-MoE-v2 on LLaMA-3 8B.

| Active Parameters | Method | # Tokens | BoolQ (32) | SciQ | PIQA | ARC-C (25) | TruthfulQA | HellaS. (10) | MMLU (5) | Average |
|---|---|---|---|---|---|---|---|---|---|---|
| | | | acc | acc-norm | acc | acc-norm | acc | acc-norm | acc | |
| 100% | LLaMA-3 8B | 15T | 83.00 | 93.20 | 78.51 | 61.86 | 51.71 | 78.79 | 67.22 | 73.33 |
| 50% | LLaMA-MoE-v2 E8A2 (Qu et al., 2024) | 7B | 74.62 | **90.60** | 69.26 | **42.83** | 45.62 | 58.95 | 37.41 | **59.61** |
| | LLaMA-MoE-v2 E8A1S1 (Qu et al., 2024) | 7B | **76.88** | 88.80 | 67.90 | 40.19 | **46.85** | 53.67 | **40.89** | 59.31 |
| | **ToMoE (Ours)** | 0.02B | 63.67 | 87.50 | **72.63** | 37.29 | 40.48 | **63.91** | 36.31 | 57.40 |

## 4.2 Language Modeling

Tab. 1 presents the perplexity results of structured pruning methods applied to LLaMA-2 models of sizes 7B and 13B on the WikiText-2 dataset, comparing various methods with 70%, 60%, and 50% of active parameters setting—corresponding to pruning ratios of 30%, 40%, and 50% for pruning, respectively. Across all pruning ratios, ToMoE consistently achieves the lowest perplexity compared to other methods, even outperforming many approaches with significantly larger numbers of active parameters. For instance, ToMoE with 50% active parameters achieves a perplexity of 8.36, which is superior to LLM-Pruner, ShortGPT, SLEB, and SliceGPT at a 30% pruning ratio. Furthermore, ToMoE with 50% active parameters surpasses ModeGPT and LLM Surgeon at a 40% pruning ratio. While the gap between ToMoE and DISP-LLM is smaller, it is still obvious at a 50% pruning ratio: ToMoE achieves a perplexity that is 1.48 points lower than DISP-LLM. ToMoE also exhibits superior performance with the LLaMA-2 13B model, maintaining a similar advantage over other methods as observed with the LLaMA-2 7B model. This demonstrates the effectiveness of ToMoE in maintaining strong language modeling performance, even with much fewer active parameters. Tab. 2 presents a comparison of our method against semi-structural pruning techniques. Our approach consistently achieves the lowest perplexity while retaining 50% of the active parameters. Moreover, the performance gap between our method and the semi-structural pruning methods is also obvious. On the LLaMA-2 7B model, SparseGPT achieves the second-best performance, with our method improving upon it by 1.81 in terms of perplexity. For the LLaMA-2 13B model, Pruner-Zero shows the second-best performance, while ToMoE further reduces the perplexity by 0.63. The comparison against semi-structural pruning methods further demonstrates the advantage of our method on the language modeling task.

Table 7: ToMoE Visualization for the last layer of the LLaMA-2 7B model with 50% active parameters

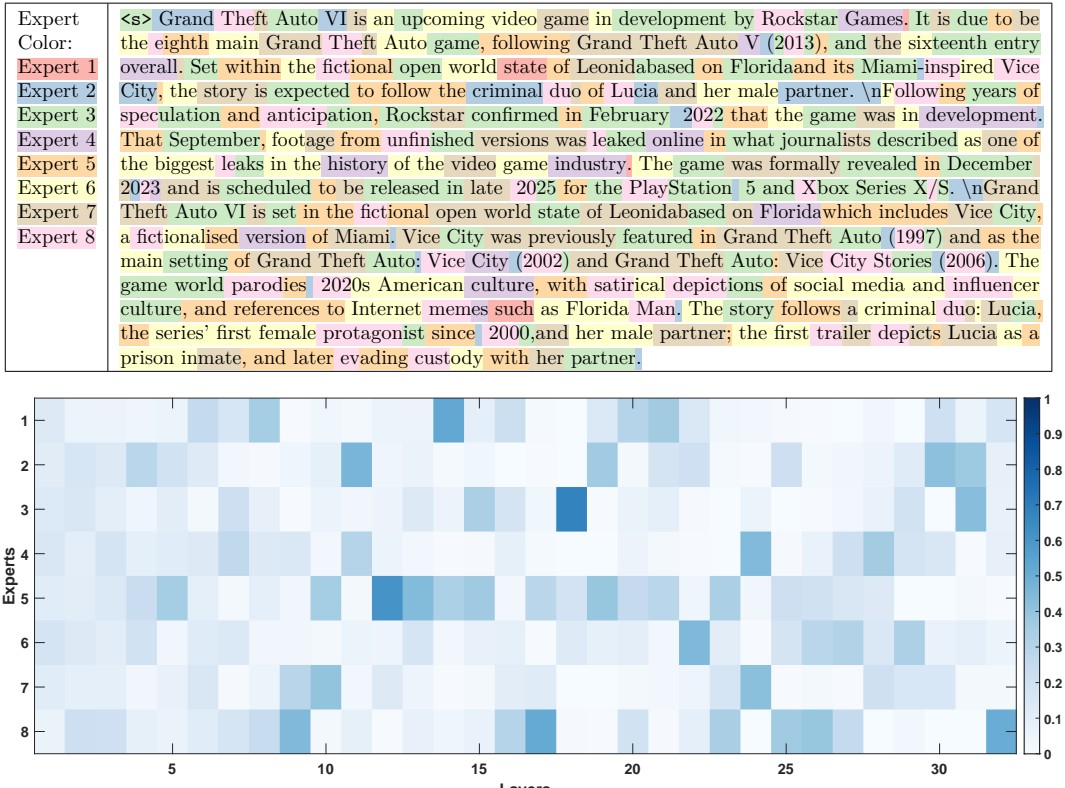

Figure 4: Experts token allocation of ToMoE for the LLaMA-3 8B model collected on the WikiText dataset.

### 4.3 Zero-Shot and Few-Shot Performance

In Tab. 3, we present the zero-shot performance of various methods on LLaMA-2 7B, LLaMA-3 8B, and Qwen-2.5 7B. Our method consistently achieves the best average performance across all models. For **LLaMA-2 7B**, compared to weaker methods like ShortGPT and SliceGPT, our approach demonstrates significant advantages (ToMoE 50%: 60.72 vs. SliceGPT 40.84 and ShortGPT 70%: 47.07). The advantage against stronger baselines is also obvious. Although ModeGPT performs closer to ToMoE, the gap remains significant. With 60% active parameters, ToMoE is 3.14 times better than ModeGPT. For **LLaMA-3 8B**, the performance advantage of ToMoE is even larger, where it reduces 5% more active parameters than ModeGPT while still achieving a 2.71 performance gain. Furthermore, when removing 15% more active parameters compared to ShortGPT and SliceGPT, ToMoE exceeds their average performance by 18.93 and 16 points, respectively. For **Qwen-2.5 7B**, ToMoE significantly outperforms DISP-LLM, consistent with previous findings on other models. We further investigate the effect of the number of experts $N$ when 40% to 50% of the parameters are active. The results indicate that increasing the number of experts to 16 is beneficial. However, further increasing $N$ to 24 provides only marginal or no improvement, likely because a too-large number of experts burdens the learning process. Thus, we recommend choosing the number of experts $N$ to be smaller than 16.

In Tab. 4, our method demonstrated superb advantages compared to existing MoE construction methods. In "+fine-tuning" setting of LLaMA-MoE, the resulting model is trained for the same number of iterations as ToMoE for updating model weights. In Tab. 5 and Tab. 6, we further compare our method with CMoE, LLaMA-MoE, and LLaMA-MoE-v2, following the experimental settings in their papers. Our approach consistently surpasses all baselines while requiring significantly fewer tokens. Notably, in Tab. 6, when compared against the fully trained LLaMA-MoE-v2, our method achieves comparable performance even without additional fine-tuning of the model weights. In summary, our method shows that learning routers and experts together is a more promising solution compared to existing works.

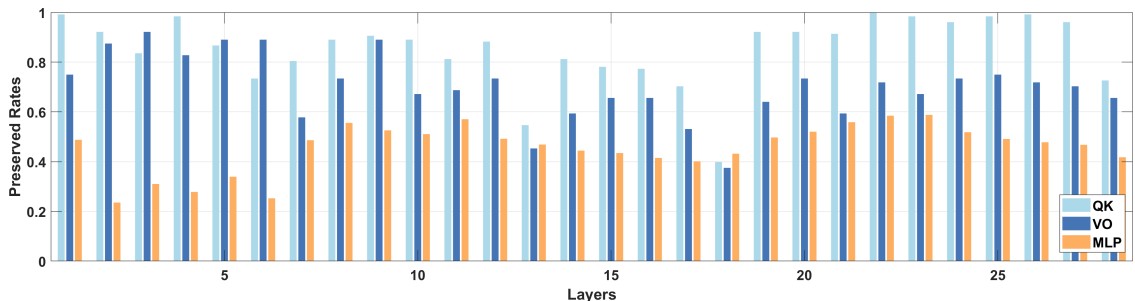

Figure 5: Model width after ToMoE for the Qwen-2.5 7B model when the number of active parameters equals 50%.

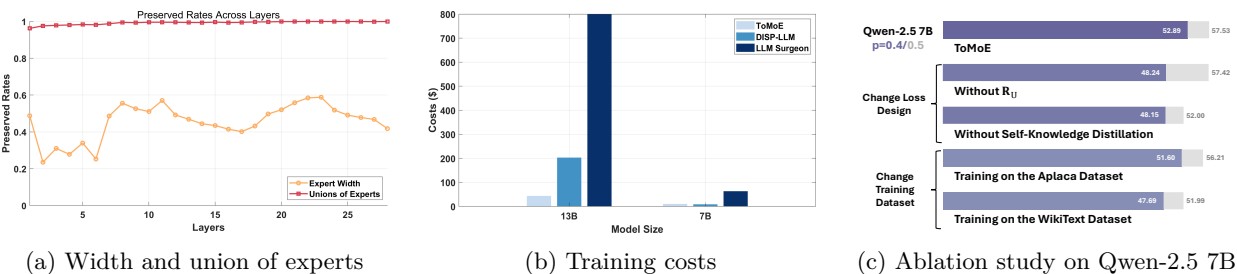

(a) Width and union of experts    (b) Training costs    (c) Ablation study on Qwen-2.5 7B

Figure 6: (a) Model width and union of experts. (b) Costs of different learning-based methods. (c) Ablation study on Qwen-2.5 7B.

## 4.4  Analysis of ToMoE

**Training Dynamics.** In Fig.3, we visualize the training dynamics under different values of $p$. Across all $p$, the knowledge distillation loss $\mathcal{L}$ (Fig.3(a)), the parameter regularization loss $\mathcal{R}_{\mathrm{P}}$, and the union of experts regularization loss $\mathcal{R}_{\mathrm{U}}$ decrease over the course of training. Notably, the parameter regularization loss quickly drops to 0 in the early stages of training, while using a smaller $p$ requires more iterations. The peak of the union of experts regularization loss increases when using smaller values of $p$, indicating that the initial solution tends to only cover a small portion of the dense model. Regarding the load balancing loss, it oscillates around 0.15, demonstrating that ToMoE maintains a relatively balanced load distribution during the training process.

**Ablation Study.** In Fig. 6c, we present the average zero-shot task performance under different settings. For $p = 0.4$ and $p = 0.5$, replacing the knowledge distillation loss with the language modeling loss significantly impacts performance. At $p = 0.4$, removing $\mathcal{R}_{\mathrm{U}}$ also results in a substantial performance drop, whereas the impact is much smaller at $p = 0.5$. We hypothesize that this difference arises because reducing $p$ makes the learning process more challenging. Without the guidance provided by $\mathcal{R}_{\mathrm{U}}$, the model struggles to effectively utilize the parameters of the original model. Additionally, the choice of dataset affects performance, particularly when switching to the WikiText dataset. This demonstrates that a mixing dataset is beneficial to the overall performance.

**Other Analysis. (1).** Fig. 5 presents the width of our ToMoE model for Qwen-2.5 7B, which shows the layer-wise configuration is highly non-uniform. It demonstrates that our method can flexibly set the width of different layers and operations. **(2).** Fig. 6a shows that the union of experts is close to the full model capacity, even though the width of experts across different layers is highly non-uniform, demonstrating the effectiveness of our loss design. **(3).** Fig. 6b plots the costs of different learning-based methods in terms of US dollars. ToMoE costs similarly compared to DISP-LLM with LLaMA-2 7B and 13B models, and both of them are much cheaper than LLM Surgeon. **(4).** Fig. 4 shows the token allocation across experts on the Wikitext dataset. We observe that the early and late layers exhibit relatively balanced expert utilization, while the middle layers have certain experts activated more frequently. **(5).** Finally, we visualize the expert

selection for LLaMA-2 7B in Tab. 7. We can observe that each expert aligns syntax rather than semantic meanings, resembling the observations in (Jiang et al., 2024).

## 5 Conclusion

In this paper, we propose a novel algorithm, ToMoE, for converting dense models into MoE models through dynamic structural pruning. The resulting MoE models significantly outperform state-of-the-art structural pruning methods while using similar or lower training costs compared to other learning-based pruning methods. Our findings reveal the presence of meaningful experts within the MLP layers of dense models, even without fine-tuning the model weights. ToMoE serves as a powerful tool for uncovering these experts within the original dense LLM.

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

# A  Details of trainable modules

## A.1  Module Configurations

We present the details of trainable modules in Tab. 8. In short, we project the input tokens to a low-dimensional space and add them to the output of the HyperNetwork. The inputs $z$ to the HyperNetwork are fixed random vectors of size $N \times 32$ sampled from a Normal Distribution. Except for the HyperNetwork, other individual trainable modules are created for each MHA and MLP layer. If we have $L$ blocks, then we will have $L$ $\text{Proj}_E^{\text{MHA}}$, $L$ $\text{Proj}_D^{\text{MHA}}$ with output size of $\frac{d}{H}$, $L$ $\text{Proj}_D^{\text{MHA}}$ with output size of $\frac{d}{2H}$ $L$, $\text{Proj}_D^{\text{MLP}}$ and $L$ Router layers. Notations of $d$, $H$, $d_{\text{mid}}$, and $N$ are already defined in Sec. 3.

| Module Types | Removed? | Structures |
|---|:---:|:---:|
| HyperNetwork | ✓ | Input $z \to$ Bi-GRU(32,64) |
| $\text{Proj}_E^{\text{MHA}}$ | ✗ | Linear($d$, $d_e = 128$) |
| $\text{Proj}_D^{\text{MHA}}$ | ✗ | LayerNorm($d_e = 128$)$\to$ GeLU$\to$Linear($d_e = 128$, $\frac{d}{H}$) |
| $\text{Proj}_D^{\text{MLP}}$ | ✓ | LayerNorm($d_e = 128$)$\to$ GeLU$\to$Linear($d_e = 128$, $d_{\text{mid}}$) |
| Router | ✗ | Linear($d$, $N$) |

Table 8: Detailed configuration of trainable modules.

After we complete the training of ToMoE, we do not have to preserve all modules. The embeddings from the HyperNetwork will be saved, so the HyperNetwork can be removed without impacting the model. $\text{Proj}_D^{\text{MLP}}$ brings most additional parameters, fortunately, it can also be removed. After the training of ToMoE, $\mathbf{E}$ and $\text{Proj}_D^{\text{MLP}}$ can be used to directly generate experts:

$$\mathbf{s_e} = \text{ST-GSig}(\text{Proj}_D^{\text{MLP}}(\mathbf{E})), \tag{11}$$

where ST-GSig again is the Straight-Through Gumbel-Sigmoid function. $\mathbf{s_e} \in \{0,1\}^{N \times d_{\text{mid}}}$ is the resulting binary vectors to select experts from the dense model. Once $\mathbf{s_e}$ is generated, it can be reused, and thus we no longer need $\text{Proj}_D^{\text{MLP}}$. Let $\mathbf{S_e}^i = \text{Diag}(\mathbf{s_e}^i)$, $i = 1, \cdots, N$. Similarly, we use $\hat{\mathbf{s}}_\mathbf{e}^i \in \Re^{d_{\text{mid}} \times d'_{\text{mid}}}$ to represent the actual column or row selection matrix by removing zero columns or rows, where $d'_{\text{mid}} < d_{\text{mid}}$ and it is the width of each expert. The $i$th expert can be represented as:

$$f_{\text{MLP}}^i(\mathbf{X}) = \sigma(\mathbf{X}\mathbf{W}_G \hat{\mathbf{S}}_\mathbf{e}^i) \odot (\mathbf{X}\mathbf{W}_U \hat{\mathbf{S}}_\mathbf{e}^i)\hat{\mathbf{S}}_\mathbf{e}^i \mathbf{W}_D. \tag{12}$$

After ToMoE, given the result of the routing function $\mathbf{G} = \text{ST-GSmax}(\text{Router}(\mathbf{X}))$, the MLP calculation with MoE can be written as:

$$\mathbf{Y}_t = \mathbf{G}_{t,i} f_{\text{MLP}}^i(\mathbf{X}_t), \tag{13}$$

where $\mathbf{X}_t$ is the feature map of $t$th token, and $i$ represents the index where $\mathbf{G}_{t,i} = 1$. Note that Eq. 13 is still differentiable with respect to the parameters of the Router.

Another question is how many parameters we need after introducing Top-K routing for MHA layers and Top-1 routing for MLP layers. Analytically, the additional parameters can be calculated by $L \times d \times 128 + L \times 128 \times (\frac{d}{H}) + L \times d \times N$. Let's use LLaMA-2 7B as an example, $L = 32$, $d = 4096$, $N = 8$, $H = 32$, the additional parameters are $1 \times 32 \times 4096 \times 128 + 32 \times 128 \times (128) + 32 \times 4096 \times 8 = 0.0184B$. This is equivalent to 0.27% of the total parameters of the LLaMA-2 7B model, and thus, the additional parameter is not significant compared to the original number of parameters.

## A.2  Head Dimension Pruning vs. RoPE

Rotary Position Embedding (RoPE) (Su et al., 2024) is a popular positional encoding method, and it is regularly used in LLMs like LLaMA (Touvron et al., 2023b). RoPE divides the $\frac{d}{H}$ dimensional space into $\frac{d}{2H}$ sub-spaces, and they are applied on query and key. This means that if we want to perform head dimension pruning for query and key matrices, we need to follow the sub-spaces resulting from RoPE and

Table 9: Zero-shot task performance of compressed LLaMA-2 13B and Qwen-2.5 14B.

| Model | Active Parameters | Method | ARC-e | ARC-c | PIQA | WinoG. | HellaS. | Average |
|---|---|---|---|---|---|---|---|---|
| | | | acc-norm | acc-norm | acc-norm | acc | acc-norm | |
| LLaMA-2 13B | 100% | Dense | 77.48 | 49.23 | 80.47 | 72.22 | 79.39 | 71.76 |
| | 70% | SliceGPT (Ashkboos et al., 2024) | 60.27 | 36.18 | 69.42 | 64.09 | 49.74 | 55.94 |
| | | LLM Surgeon (van der Ouderaa et al., 2024) | 69.74 | 40.27 | 76.50 | 68.67 | 71.52 | 65.34 |
| | | DISP-LLM (Gao et al., 2024) | 63.80 | 39.42 | 74.43 | 66.85 | 70.86 | 63.07 |
| | | DISP-LLM Alpaca (Gao et al., 2024) | 68.98 | 44.28 | 77.31 | 67.32 | 68.98 | 65.59 |
| | | MoDeGPT-Alpaca (Lin et al., 2024) | 70.24 | 41.47 | 77.15 | 71.27 | 71.84 | 66.39 |
| | | **ToMoE (Ours)** | **74.58** | **45.14** | **77.97** | **75.44** | **68.19** | **68.26** |
| | 60% | SliceGPT (Ashkboos et al., 2024) | 48.99 | 32.51 | 63.17 | 56.75 | 39.85 | 48.25 |
| | | LLM Surgeon (van der Ouderaa et al., 2024) | 63.80 | 37.12 | 73.16 | 65.75 | 65.64 | 60.94 |
| | | DISP-LLM (Gao et al., 2024) | 62.67 | 35.63 | 73.39 | 62.67 | 65.86 | 60.04 |
| | | DISP-LLM Alpaca (Gao et al., 2024) | 66.79 | 42.75 | 75.30 | 64.25 | 67.52 | 63.32 |
| | | MoDeGPT-Alpaca (Lin et al., 2024) | 63.72 | 38.82 | 71.87 | 66.30 | 62.10 | 60.56 |
| | | **ToMoE (Ours)** | **67.63** | **41.81** | **75.79** | **66.38** | **73.41** | **65.00** |
| | 50% | LLM Surgeon (van der Ouderaa et al., 2024) | 56.19 | 37.12 | 68.87 | 63.22 | 56.19 | 56.32 |
| | | DISP-LLM (Gao et al., 2024) | 58.27 | 36.87 | 68.67 | 59.27 | 57.18 | 54.50 |
| | | DISP-LLM Alpaca (Gao et al., 2024) | 55.72 | 37.54 | 72.20 | 59.59 | 62.39 | 57.49 |
| | | **ToMoE (Ours)** | **63.51** | **37.97** | **73.29** | **62.59** | **68.30** | **61.13** |
| Qwen-2.5 14B | 100% | Dense | 79.42 | 59.13 | 83.09 | 74.98 | 82.10 | 75.74 |
| | 50% | DISP-LLM (Gao et al., 2024) | 65.87 | 37.99 | 72.20 | 58.51 | 60.63 | 59.04 |
| | | **ToMoE (Ours)** | **66.37** | **39.16** | **74.59** | **60.69** | **65.57** | **61.28** |
| | 40% | DISP-LLM (Gao et al., 2024) | 56.36 | 32.62 | 70.08 | 54.22 | 51.50 | 52.96 |
| | | **ToMoE (Ours)** | **59.85** | **34.73** | **71.49** | **57.62** | **56.60** | **56.06** |

make these two sub-spaces share the same pruning mask $\mathbf{s}_0$, and the final pruning mask for query and key is $\mathbf{s}'_0 = [\mathbf{s}_{0[1:\frac{d}{2H}]}, \mathbf{s}_{0[1:\frac{d}{2H}]}]$, and clearly the size of $\mathbf{s}_{0[1:\frac{d}{2H}]}$ is $\frac{d}{2H}$. In short, we simply select the first half of elements from $\mathbf{s}_0$ and repeat it twice to make the final pruning decision. We also found that applying dynamic pruning for query and key matrices along the head dimension is difficult and unreasonable since different tokens may have different positions after pruning. It becomes a problem when calculating the inner product between the query and key matrices given different tokens.

By applying head dimension pruning, our method also does not need to be modified when facing different attention mechanisms like GQA (Grouped-Query Attention) (Ainslie et al., 2023) and MQA (Multi-Query Attention) (Shazeer, 2019).

### A.3 Details of Gumbel-Softmax and Gumbel-Sigmoid

The Gumbel-Softmax function (Jang et al., 2016) allows for differentiable sampling from a categorical distribution. Given logits $\mathbf{x}$, the Gumbel-Softmax sample $\mathbf{y}$ is computed as:

$$\mathbf{y} = \text{softmax}\left(\frac{\mathbf{x} + \mathbf{g}}{\tau}\right),$$

where each element of $\mathbf{g}$ is drawn from Gumbel$(0, 1)$, and $\tau$ is the temperature parameter that controls the smoothness of the distribution. Combining Gumbel-Softmax with the Straight-Through gradient Estimator (Bengio et al., 2013), we have the following equation:

$$\text{ST-GSmax}(\mathbf{x}) = \text{one-hot}\left(\arg\max_{i \in D}\left[\frac{x_i + g_i}{\tau}\right]\right) \tag{14}$$

where $D = \{1, 2, \cdots, N\}$, $N$ again is the number of experts in our setting, and one-hot will assign 1 corresponding to the position of the maximum value in $\frac{\mathbf{x} + \mathbf{g}}{\tau}$ and assign 0 to other positions.

Table 10: Zero-shot task performance of the compressed Phi-2.

| Active Parameters | Method | ARC-e | ARC-c | PIQA | WinoG. | HellaS. | Avg |
|---|---|---|---|---|---|---|---|
| | | acc-norm | acc-norm | acc-norm | acc | acc-norm | |
| 100% | Dense | 78.24 | 54.01 | 79.11 | 75.61 | 73.86 | 72.17 |
| 80% | SliceGPT (Ashkboos et al., 2024) | 58.00 | 35.32 | 71.87 | 67.80 | 57.76 | 58.15 |
| | +Fine-tuning | 56.61 | 38.91 | 71.27 | 67.17 | 54.86 | 57.76 |
| | DISP-LLM (Gao et al., 2024) | 68.18 | **44.11** | 74.86 | 67.09 | **62.93** | 63.43 |
| 75% | SliceGPT (Ashkboos et al., 2024) | 53.70 | 31.66 | 69.21 | 65.35 | 52.40 | 54.46 |
| | +Fine-tuning | 52.78 | 35.49 | 69.91 | 65.19 | 52.48 | 55.17 |
| | DISP-LLM (Gao et al., 2024) | 65.93 | 43.34 | 74.27 | 65.11 | 59.95 | 61.72 |
| 70% | SliceGPT (Ashkboos et al., 2024) | 53.03 | 30.29 | 65.94 | 63.14 | 47.56 | 51.99 |
| | +Fine-tuning | 46.38 | 32.68 | 66.16 | 63.54 | 49.72 | 51.70 |
| | DISP-LLM (Gao et al., 2024) | 63.59 | 38.48 | 73.34 | 65.19 | 54.43 | 59.00 |
| | **ToMoE (Ours)** | **70.79** | 43.86 | **77.09** | **66.38** | 62.68 | **64.16** |

The Gumbel-Sigmoid function is a special case of the Gumbel-Softmax function, designed for binary distributions. Given logits $\mathbf{x}$, the Gumbel-Sigmoid sample $\mathbf{y}$ is computed as:

$$\mathbf{y} = \text{sigmoid}\left(\frac{\mathbf{x} + \mathbf{g}}{\tau}\right),$$

where $\mathbf{g}$ is sampled from Gumbel$(0, 1)$ and $\tau$ again is the temperature parameter. Combining with the Straight-Through gradient Estimator, we have the following equation:

$$\text{ST-GSig}(\mathbf{x}) = \text{round}(\text{sigmoid}\left(\frac{\mathbf{x} + \mathbf{g} + b}{\tau}\right)), \tag{15}$$

where $b$ is a constant bias in our implementation and it ensures that all experts start from the whole model, round$(\cdot)$ will round the input values to the nearest integer, and in our case, it rounds inputs to 0 or 1. For all experiments, we set $b = 3.0$ in Eq. 15, and we set $\tau = 0.4$ for Eq. 14 and Eq. 15.

Listing 1: Pseudo-code for self-knowledge distillation.

```python
with torch.no_grad():
    # Disable trainable modules for ToMoE
    helper.set_module_status(model, False)

    # Get logits from the teacher (original model)
    teacher_output = model(inputs)
    teacher_logits = teacher_output.logits

    # Enable trainable modules for ToMoE
    helper.set_module_status(model, True)

# Get logits from the student model from ToMoE learning
    process
model_output = model(inputs)
logits = model_output.logits
```

# B More Details of the Loss Design

## B.1 Implementation of the Self-Knowledge Distillation

During the ToMoE learning process, we freeze the parameters of the original model. This approach offers the additional benefit of enabling self-knowledge distillation without the need to load an extra model. In

Table 11: ToMoE Visualization of LLaMA-2 7B with 50% active parameters

| Expert Color | Expert 1 Expert 2 Expert 3 Expert 4 Expert 5 Expert 6 Expert 7 Expert 8 |
|---|---|
| MLP 1 | \<s\> Homarus gammarus, known as the European lobster or common lobster, is a species of clawed lobster from the eastern Atlantic Ocean, Mediterranean Sea and parts of the Black Sea. It is closely related to the American lobster, H. americanus. It may grow to a length of 60 cm (24 in) and a mass of 6 kilograms (13 lb), and bears a conspicuous pair of claws. In life, the lobsters are blue, only becoming 'lobster red' on cooking. Mating occurs in the summer, producing eggs which are carried by the females for up to a year before hatching into planktonic larvae. Homarus gammarus is a highly esteemed food, and is widely caught using lobster pots, mostly around the British Isles. |
| MLP 16 | \<s\> Homarus gammarus, known as the European lobster or common lobster, is a species of clawed lobster from the eastern Atlantic Ocean, Mediterranean Sea and parts of the Black Sea. It is closely related to the American lobster, H. americanus. It may grow to a length of 60 cm (24 in) and a mass of 6 kilograms (13 lb), and bears a conspicuous pair of claws. In life, the lobsters are blue, only becoming 'lobster red' on cooking. Mating occurs in the summer, producing eggs which are carried by the females for up to a year before hatching into planktonic larvae. Homarus gammarus is a highly esteemed food, and is widely caught using lobster pots, mostly around the British Isles. |
| MLP 32 | \<s\> Homarus gammarus, known as the European lobster or common lobster, is a species of clawed lobster from the eastern Atlantic Ocean, Mediterranean Sea and parts of the Black Sea. It is closely related to the American lobster, H. americanus. It may grow to a length of 60 cm (24 in) and a mass of 6 kilograms (13 lb), and bears a conspicuous pair of claws. In life, the lobsters are blue, only becoming 'lobster red' on cooking. Mating occurs in the summer, producing eggs which are carried by the females for up to a year before hatching into planktonic larvae. Homarus gammarus is a highly esteemed food, and is widely caught using lobster pots, mostly around the British Isles. |

Table 12: Zero-shot task performance of compressed LLaMA-2 7B with more settings.

| Active Parameters | Method | ARC-e | ARC-c | PIQA | WinoG. | HellaS. | Average |
|---|---|---|---|---|---|---|---|
| | | acc-norm | acc-norm | acc-norm | acc | acc-norm | |
| 100% | Dense | 74.58 | 46.25 | 79.11 | 69.06 | 75.99 | 69.00 |
| 70% | ShortGPT (Men et al., 2024) | 48.65 | 32.85 | 64.31 | 64.33 | 56.13 | 53.25 |
| | SliceGPT (Ashkboos et al., 2024) | 58.88 | 33.36 | 68.55 | 58.01 | 49.86 | 53.73 |
| | LLM Surgeon (van der Ouderaa et al., 2024) | 63.09 | 36.69 | 73.56 | 61.09 | 60.72 | 59.03 |
| | DISP-LLM (Gao et al., 2024) | 59.81 | 33.19 | 71.82 | 62.27 | 63.43 | 58.10 |
| | DISP-LLM Alpaca (Gao et al., 2024) | 60.10 | 37.03 | 73.72 | 63.93 | 62.87 | 59.53 |
| | ModeGPT (Lin et al., 2024) | 63.26 | 38.73 | 70.40 | 67.32 | 63.26 | 60.78 |
| | ModeGPT-Alpaca (Lin et al., 2024) | 65.49 | 39.16 | 73.34 | 66.22 | 65.90 | 62.02 |
| 60% | ShortGPT (Men et al., 2024) | 41.16 | 29.94 | 60.12 | 60.46 | 43.67 | 47.07 |
| | SliceGPT (Ashkboos et al., 2024) | 36.49 | 24.57 | 54.90 | 53.43 | 34.80 | 40.84 |
| | LLM Surgeon (van der Ouderaa et al., 2024) | 52.31 | 30.29 | 69.26 | 54.38 | 48.04 | 50.86 |
| | ModeGPT (Lin et al., 2024) | 49.45 | 30.03 | 64.96 | 61.96 | 53.01 | 51.88 |
| | ModeGPT-Alpaca (Lin et al., 2024) | 59.76 | 34.73 | 70.35 | **64.40** | 58.63 | 57.58 |
| | **ToMoE (Ours)** | **63.64** | **38.74** | **72.85** | 62.51 | **65.84** | **60.72** |
| 50% | LLM Surgeon (van der Ouderaa et al., 2024) | 44.91 | 26.28 | 64.36 | 52.57 | 40.29 | 45.68 |
| | DISP-LLM (Gao et al., 2024) | 43.06 | 25.85 | 63.93 | 54.54 | 63.43 | 46.72 |
| | DISP-LLM Alpaca (Gao et al., 2024) | 51.14 | 30.20 | 68.34 | 56.20 | 49.35 | 51.05 |
| | **ToMoE (Ours)** | **56.65** | **33.87** | **71.00** | **58.56** | **60.26** | **56.07** |

Lst. 1, we present the pseudo-code for the self-knowledge distillation process. In summary, we first disable the trainable modules associated with ToMoE and compute the output logits from the original model. Next, we re-enable the trainable modules for ToMoE and perform a regular forward pass. The logits from the original model are then used to guide the learning of ToMoE.

## B.2 Efficient Implementation of $\mathcal{R}_{\mathbf{u}}$

Recall from Eq. 6 that the union regularization for MLP and MHA layers is defined as:

$$\mathcal{R}_{\mathbf{u}} = \bigcup_{i=1}^{T} \mathbf{s}_i = 1 - \prod_{i=1}^{T} (1 - \mathbf{s}_i).$$

For MLP layers, this equation incurs high computational costs since $\mathbf{s} \in \mathbb{R}^{T \times d_{\mathrm{mid}}}$, whereas for MHA layers, the cost is significantly lower because $\frac{d}{H} \ll d_{\mathrm{mid}}$. To simplify Eq. 6, note that all $\mathbf{s}_i$ ($i = 1, \ldots, T$) are derived from $N$ experts. Using embeddings from the hypernetwork, we calculate the configuration of $N$ experts as:

$$\mathbf{s_e} = \text{ST-GSig}(\text{Proj}_{\mathrm{D}}^{\mathrm{MLP}}(\mathbf{E})),$$

and substitute $\mathbf{s_e}$ into Eq. 6:

$$\mathcal{R}_{\mathbf{u}}^{\mathrm{MLP}} = \bigcup_{i=1}^{N} \mathbf{s}_{\mathbf{e}}^{i} = 1 - \prod_{i=1}^{N}(1 - \mathbf{s}_{\mathbf{e}}^{i}). \tag{16}$$

This reduces computation by a factor of $\frac{T}{N}$. For example, in LLaMA-2, the computational cost is reduced by $\frac{2048}{8} = 256$ times.

Table 13: Ablation study on design choices of ToMoE and the impact of temperature $\tau$ on performance.

| Settings | ARC-e | ARC-c | PIQA | WinoG. | HellaS. | Avg |
|---|---|---|---|---|---|---|
| Local Emb | 55.22 | 32.94 | 66.32 | 53.12 | 57.85 | 53.09 |
| Head Pruning | 45.12 | 25.85 | 64.82 | 49.49 | 39.65 | 44.99 |
| w/o VO Routing | 55.51 | 32.34 | 70.40 | 57.14 | 59.12 | 54.90 |
| $\tau = 0.3$ | 57.28 | 33.53 | 70.35 | 57.54 | 60.22 | 55.78 |
| $\tau = 0.4$ | 56.65 | 33.87 | 71.00 | 58.56 | 60.26 | 56.07 |
| $\tau = 0.5$ | 55.89 | 33.36 | 71.00 | 56.75 | 59.31 | 55.16 |
| ToMoE | 56.65 | 33.87 | 71.00 | 58.56 | 60.26 | 56.07 |

### B.3 Equivalence of MoE and pseudo-MoE

One major challenge when training MoE models is maintaining an appropriate expert capacity, defined as the number of tokens each expert processes (Fedus et al., 2022). This is typically addressed using a load balancing loss. Without this loss, some experts may become overloaded while others remain underutilized, leading to bottlenecks where a few experts dominate the computation.

Although ToMoE also requires load balancing loss, the potential overhead introduced by load balancing is mitigated by the pseudo-MoE approach after ToMoE. After applying ToMoE, the resulting model can be trained using pseudo-MoE, which resembles the training of a dense model. This is straightforward to implement as follows:

$$f_{\mathrm{MLP}}(\mathbf{X}) = \sigma(\mathbf{X}\mathbf{W}_G)\mathbf{S} \odot (\mathbf{X}\mathbf{W}_U\mathbf{S})\mathbf{S}\mathbf{W}_D, \tag{17}$$

where $\mathbf{S}_i$ in $\mathbf{S}$ represents the routed expert from $\mathbf{S}_e$ in Eq. 11, as determined by the router. The pseudo-MoE is useful when the active number of parameters is relatively large. In such cases, pseudo-MoE training can be more time-efficient than conventional MoE training.

## C More Implementation Details

During training the modules of ToMoE, we use AdamW optimizer to optimize it with a constant learning rate $10^{-3}$ and weight decay 0.05. For different models, we always set the mini-batchsize to 1 on each GPU. For LLaMA-2 7B, and Qwen-2.5 7B models, we use 2 NVIDIA A100 GPUs, For LLaMA-3 8B, we use 3 NVIDIA A100 GPUs. For LLaMA-2 13B and Qwen-2.5 14B models, we use 4 NVIDIA A100 GPUs. For all the rest models, we use 1 NVIDIA A100 GPU. We set $p = \{0.6, 0.5, 0.4, 0.3\}$ when the ratios of active parameters equals to $\{40\%, 50\%, 60\%, 70\%\}$.

For the Alpaca dataset [1], we use the 'text' column within the dataset, which combines the columns of 'instruction' and 'output'. For the Code Alpaca dataset [2], we combine the 'instruction', 'input', and 'output' columns as one training sample.

---

[1]https://huggingface.co/datasets/tatsu-lab/alpaca
[2]https://github.com/sahil280114/codealpaca

Table 14: ToMoE Visualization of LLaMA-2 7B with 50% active parameters (continued).

| Expert Color | Expert 1 Expert 2 Expert 3 Expert 4 Expert 5 Expert 6 Expert 7 Expert 8 |
|---|---|
| MLP 1 | \<s\> Find the equation of the line passing through the points (3, 5) and (7, 9) using the slope-intercept form of a linear equation. To find the equation of the line passing through the points (3, 5) and (7, 9) using the slope-intercept form (y = mx + b), we first need to find the slope (m) and the y-intercept (b). 1. Find the slope (m): m = (y2 - y1) / (x2 - x1) m = (9 - 5) / (7 - 3) m = 4 / 4 m = 1 2. Use one of the points to find the y-intercept (b). We'll use (3, 5): y = mx + b 5 = 1 * 3 + b 5 = 3 + b b = 2 3. Write the equation in slope-intercept form: y = mx + b y = 1x + 2 y = x + 2 The equation of the line passing through the points (3, 5) and (7, 9) is y = x + 2. |
| MLP 16 | \<s\> Find the equation of the line passing through the points (3, 5) and (7, 9) using the slope-intercept form of a linear equation. To find the equation of the line passing through the points (3, 5) and (7, 9) using the slope-intercept form (y = mx + b), we first need to find the slope (m) and the y-intercept (b). 1. Find the slope (m): m = (y2 - y1) / (x2 - x1) m = (9 - 5) / (7 - 3) m = 4 / 4 m = 1 2. Use one of the points to find the y-intercept (b). We'll use (3, 5): y = mx + b 5 = 1 * 3 + b 5 = 3 + b b = 2 3. Write the equation in slope-intercept form: y = mx + b y = 1x + 2 y = x + 2 The equation of the line passing through the points (3, 5) and (7, 9) is y = x + 2. |
| MLP 32 | \<s\> Find the equation of the line passing through the points (3, 5) and (7, 9) using the slope-intercept form of a linear equation. To find the equation of the line passing through the points (3, 5) and (7, 9) using the slope-intercept form (y = mx + b), we first need to find the slope (m) and the y-intercept (b). 1. Find the slope (m): m = (y2 - y1) / (x2 - x1) m = (9 - 5) / (7 - 3) m = 4 / 4 m = 1 2. Use one of the points to find the y-intercept (b). We'll use (3, 5): y = mx + b 5 = 1 * 3 + b 5 = 3 + b b = 2 3. Write the equation in slope-intercept form: y = mx + b y = 1x + 2 y = x + 2 The equation of the line passing through the points (3, 5) and (7, 9) is y = x + 2. |

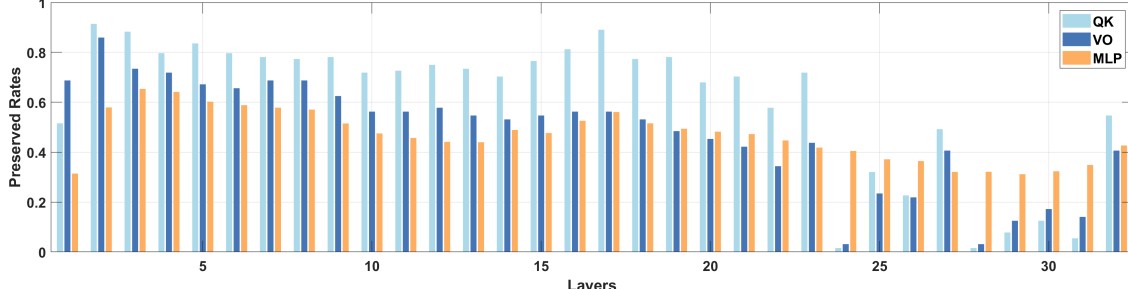

Figure 7: Model width after ToMoE for the LLaMA-2 7B model when the number of active parameters equals 50%.

## D  More Experimental Results

| Model | Active Parameters | Batch Size 512 | Batch Size 1024 | Batch Size 1536 |
|---|---|---|---|---|
| LLaMA-2 7B | 100% | 1781.76 tokens/s | 1805.80 tokens/s | 1858.45 tokens/s |
| LLaMA-MoE E8A2 (Zhu et al., 2024) | 50% | 2685.35 tokens/s | 2796.07 tokens/s | 2887.98 tokens/s |
| ToMoE (ours) | | 2360.38 tokens/s | 2702.12 tokens/s | 2919.45 tokens/s |

Table 15: Inference throughput (tokens per second) under different mini-batchsizes.

In Tab. 9 and Tab. 10, we present the zero-shot performance of various methods on LLaMA-2 13B, Qwen-2.5 14B, and Phi-2 models. From Tab. 9, it is evident that ToMoE consistently outperforms other methods. Compared to 7B or 8B models, the performance gap between our method and other approaches is smaller, which also holds for the differences between baseline methods. This is likely due to the larger model sizes. Table 12 presents the zero-shot performance of the LLaMA-2 7B model across more baselines and active parameters. As shown in the table, ToMoE consistently achieves significantly better performance than all competing methods.

On the LLaMA-2 13B model, ToMoE surpasses structural pruning methods even with smaller compression rates. For instance, ToMoE with 50% active parameters performs better than MoDeGPT and LLM Surgeon

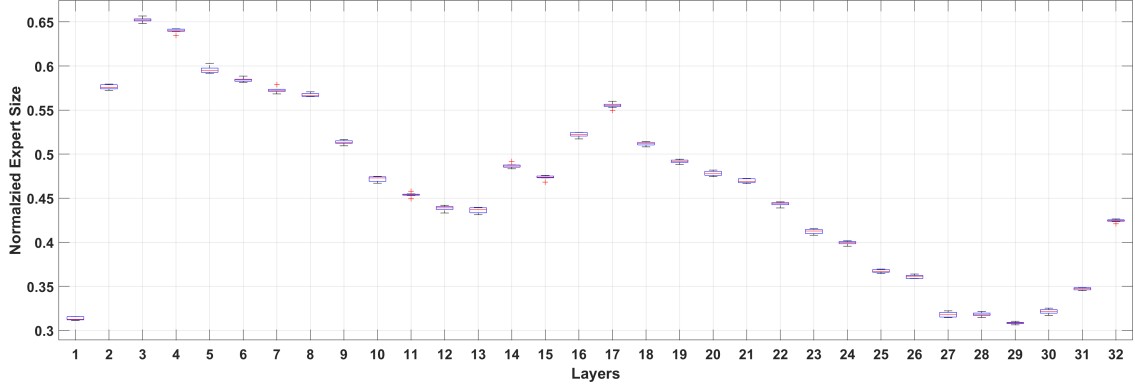

Figure 8: Box plot of widths across different experts for the LLaMA-2 7B model when the number of active parameters equals 50%.

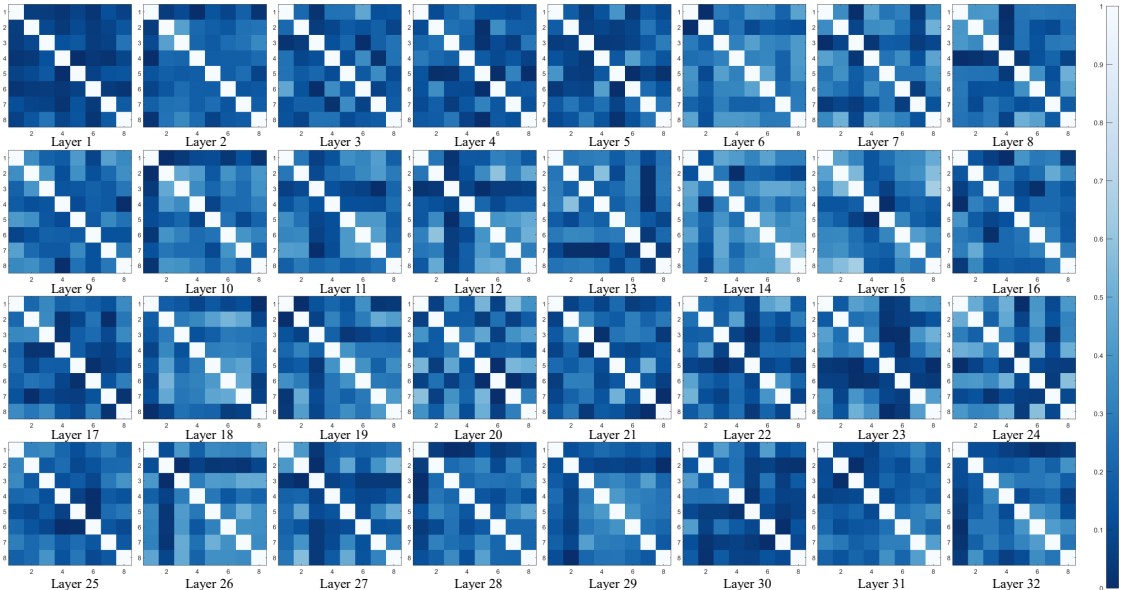

Figure 9: The similarity of different experts from different layers of ToMoE of the LLaMA-2 7B model.

with a 40% compression rate. The performance gap becomes even more obvious when comparing methods with the same number of active parameters. Similarly, from Tab. 10, ToMoE demonstrates superior performance compared to SliceGPT and DISP-LLM. Specifically, ToMoE with 70% active parameters achieves better results than all three compression levels of SliceGPT and DISP-LLM.

In Tab. 15, we report the inference throughput (measured in tokens per second) of different models under varying batch sizes. Compared to the dense LLaMA-2 7B baseline, both converted MoE models achieve higher throughput due to reduced active parameters. Our resulting model has a similar throughput to LLaMA-MoE when the batch size is large enough.

In Fig. 10, we present the accuracy–parameter trade-off on the more challenging MMLU dataset with the LLaMA-3 8B model. The results show that our method can still provide meaningful results when activating only 50% of the model parameters.

In Fig. 7, we illustrate the width of ToMoE for the LLaMA-2 7B model. A highly non-uniform pattern emerges in the allocation of active parameters, indicating

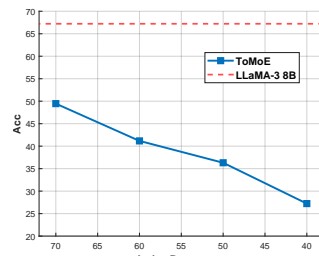

Figure 10: MMLU accuracy vs. active parameters.

that ToMoE can effectively determine the ideal distribution of active parameters, even when the allocation is highly non-uniform.

In Tab. 13, we present an ablation study to analyze several design choices in ToMoE, focusing on architectural components and the effect of the temperature parameter $\tau$. Although ToMoE applies contextual sparsity to the value and output (VO) projections within the attention layer, their contribution to overall performance is relatively minor due to the small head dimension (128, in the case of LLaMA-2 7B). To validate this, we disable dynamic attention sparsity and instead apply only static structural pruning to the attention layer (denoted as "w/o VO routing"). This leads to only a modest performance drop of about 1% using 50% active parameters.

We also examine the role of global expert embeddings with GRU in conveying cross-layer architectural information. Specifically, we compare the default global expert embedding with a local-only variant ("local emb"), where expert embeddings are used only in MLP layers and removed from attention layers. Results show a slight decrease in performance, suggesting that global expert embeddings contribute to better coordination across layers.

Additionally, we evaluate the sensitivity of ToMoE to the temperature $\tau$ in the routing mechanism. The results with $\tau \in \{0.3, 0.4, 0.5\}$ show that performance remains relatively stable, indicating robustness to the choice of $\tau$ within a reasonable range.

Finally, we explore head pruning in the early stages of ToMoE development. However, this approach yielded significantly lower performance. This may be due to the distortion of attention feature maps when heads are removed, which makes it more difficult to train effective MLP experts.

These results highlight the effectiveness of ToMoE in preserving the capacity of LLMs compared to structural pruning methods. Additionally, they demonstrate that ToMoE performs robustly across various scales and types of LLMs.

## E  Visualization of Experts

In this section, we analyze the properties of the experts produced by our method. Tab. 11 and Tab. 14 present visualizations of the routed tokens among experts across different layers and input texts.

In Tab. 11, we observe no distinct patterns in the allocation of tokens to specific experts, which aligns with our observations in Tab. 7. An interesting trend emerges when comparing layers: the first layer exhibits a more diverse token distribution, while subsequent layers prefer to assign continuous tokens to the same expert. Tab. 14 focuses on inputs related to a math problem. Unlike the visualization in Tab. 11, the MoE routing for the math problem reveals clearer semantic patterns. For instance, Expert 2 in MLP 16 is predominantly activated by numbers and mathematical notations, and a similar behavior is observed for Expert 8 in MLP 32. This suggests that the experts in ToMoE may encode more distinct semantic meanings compared to MoE models trained from scratch. Further investigation is required to fully understand the precise semantic roles of ToMoE experts.

In Fig. 8, we present a box plot showing the expert sizes across different layers. The figure reveals that the maximum and minimum expert sizes are closely aligned across layers. This outcome is a direct result of applying constraints from Eq. 8 and Eq. 9, as well as only penalizing the largest expert in Eq. 8. During training, minimizing the task loss (self-knowledge distillation loss) encourages experts to grow in size. Consequently, smaller experts do not remain small due to the task loss and they are not penalized by the parameter regularization loss. This iterative process leads to all experts eventually converging to similar sizes. After completing the ToMoE training process, we adjust the width of all experts to **match the maximum size among them**. This ensures uniform computational cost across all experts.

In Fig. 9, we present a visualization of the similarity between different experts across all layers of the LLaMA-2 7B model. Within the same layer, experts generally exhibit comparable similarity values, indicating that while the experts share the same size, their weights remain distinct. Notably, certain layers, such as layer 1 and layer 30, show lower similarity values. This observation aligns with expectations, as the expert sizes in these layers are smaller.

# F   Design Choice for the MHA

Ideally, to achieve maximum flexibility, one might consider applying dynamic pruning to all projection matrices in the MHA layer, including the query ($W_Q$), key ($W_K$), value ($W_V$), and output ($W_O$) matrices. However, there is a fundamental limitation when attempting to apply dynamic pruning along the head dimension for the query and key matrices.

Suppose we generate pruning masks $S_t \in \{0,1\}^d$ at each time step $t$ based on the input $X_t \in \mathbb{R}^{1 \times d}$, and consider two distinct time steps, $a$ and $b$. For the $i$-th attention head, the attention score between queries and keys is influenced by the pruning masks. Specifically, the effective attention score between the $a$-th query and the $a$-th key is given by:

$$e(X_a W_{Q,i}) S_a S_a^\top e(X_a W_{K,i})^\top,$$

while the attention score between the $a$-th query and the $b$-th key is:

$$e(X_a W_{Q,i}) S_a S_b^\top e(X_b W_{K,i})^\top.$$

The *effective width*—that is, the dimensionality over which attention is computed—between the $a$-th query and the $a$-th key is $\|S_a S_a^\top\|_0 = \sum S_a = K$, assuming the mask has exactly $K$ active elements. However, for cross-position pairs like $(a, b)$, the effective width becomes $\|S_a S_b^\top\|_0 = \|S_a \odot S_b\|_0 \leq \min(\sum S_a, \sum S_b) = K$. The equality holds only when $S_a = S_b$, which generally does not hold for arbitrary $a \neq b$.

This observation implies that dynamically pruned query and key matrices fail to fully utilize the allocated capacity $K$ unless the pruning masks are identical across all positions. Moreover, the variability of the effective width across different query-key pairs introduces instability and inconsistent capacity utilization, making this approach less favorable compared to static pruning for the query and key matrices.

In contrast, dynamic pruning does not encounter this issue when applied to the value and output matrices, as these are not involved in pairwise comparisons like the query-key dot products. Therefore, we adopt dynamic pruning only for the value and output projections, while keeping the query and key projections pruned statically to maintain stable and full-capacity attention computation.

