# OpenReview forum: "ToMoE: Converting Dense Large Language Models to Mixture-of-Experts through Dynamic Structural Pruning"
_TMLR — Accepted by TMLR_

### Review · Reviewer_o8BZ · 2025-10-21

**Summary Of Contributions:**

Summary

This paper provides a new aspect to dynamically prune dense models to MoE models, which could utilize the scaling and large model capacity properties of MoE to maximize the model performance. ToMoE performs pruning on both attentions and FFNs, and converts the model to a sparse one effectively. Experiments are conducted under different activation settings and model series, and the performance indicate its effectiveness.

Strengths

- ToMoE combines the structural pruning concept to converting dense models into dynamic sparse MoEs, which is a novel contribution.
- The training process is highly efficient. Comparing to continual pre-training and structural pruning, ToMoE could reach relative high results with minimal training costs.
- The experimental results are promising, and it achieves superior performance compared to structural pruning, semi-structural pruning, and continual pre-training.
- The ablation studies are insightful, providing sufficient results to understanding the training dynamics.

Weaknesses

- The main results do not include harder tasks. According to previous dense-to-MoE baselines, MMLU seems to be challenging and have a significant performance drop after model conversion. It would be better to evaluate models in MMLU, BBH, and GSM8K benchmarks.
- Table 4 is a case study (expert 1 is less utilized), which is less quantitively useful. Please change it to a expert load heatmap with layers as x-axis and experts as y-axis.
- The definition and initialization methods of the router are missing, and it would be better to explain them in the main content.
- It is not clear if this method could bring efficiency improvements. Although the number of activated parameters could be reduced by pruning, such a dynamic structural pruning may not be more efficient due to those selection matrices (masks). Could you provide an efficiency analysis (especially on the inference stage)?
- `without fine-tuning` in abstract and introduction may be over claimed, and please consider revise this statement since ToMoE still has to train routers and selection matrices.

**Audience:**

Yes

**Audience Explanation:**

Yes. MoE, model pruning, and model conversion communities may find this paper useful for their research or take it as a baseline.

**Claims And Evidence:**

Yes

**Claims Explanation:**

The most important claim is that structural pruning would remove necessary model parameters and lead to performance decline. Based on the overall experimental results, I think ToMoE has shown the potential to surpass those baselines.

However, the `without fine-tuning` claim may be over stated and should be revised since ToMoE still needs router tuning.

**Requested Changes:**

- WikiText, WikiText-2, Alpaca, CodeAlpaca are not properly cited or referred.
- The conclusions in Table 11 and the others are insightful, please consider summarize some takeaways at the end of the introduction section to strengthen the overall contributions.
- Formats
    - LLama → LLaMA
    - For vectors, please omit $\times 1$ in shapes, e.g. \textbf{1}_{d_{mid}} \in \mathbb{R}^{d_{mid} \times 1} → \textbf{1}_{d_{mid}} \in \mathbb{R}^{d_{mid}}
    - than → then before Eq. 7
    - missing comma at the end of equations since there are `where` following after
    - some symbols are conflicting, e.g. the `f` is both a regression function and a model

---

> ### Author Response · Authors · 2025-11-09
> **Response to Reviewer o8BZ**
>
> We want to thank Reviewer o8BZ for giving constructive feedback regarding our paper. We provided the following response below and **in the revised version of our paper**.
>
> ### 1. The main results do not include harder tasks.
>
> We further evaluate the converted model on 5-shot MMLU and compared it with LLaMA-MoE-v2, as shown in Table 6 in our revision. LLaMA-MoE-v2 is fine-tuned for 7B tokens. Our method, even without fine-tuning on model weights, still achieves close MMLU performance compared to LLaMA-MoE-v2.
>
> ### 2. Expert load heatmap.
>
> We provided the expert load heatmap in Figure 4 of our revised version.
>
> ### 3. The definition and initialization methods of the router are missing.
>
> We have added a sentence clarifying the definition of the router in Section 3.2, and a more detailed description is provided in Table 8 of the appendix. The router is implemented as a simple linear layer with the default initialization from PyTorch; therefore, we did not include it in the paper.
>
> ### 4. It is not clear if this method could bring efficiency improvements.
>
> We added the comparison of the throughput of ToMoE, LLaMA-MoE, and LLaMA-2 7B model in Table 15 of the appendix. {ToMoE} achieves throughput improvements, particularly when the mini-batch size is large.
>
> ### 5. Without fine-tuning in abstract and introduction may be over claimed.
>
> We have revised them to a more precise form: “without fine-tuning of the model weights.” And we also pointed out that our method requires fine-tuning/training the router.
>
> ### 6. WikiText, WikiText-2, Alpaca, CodeAlpaca are not properly cited or referred.
>
> We added references for all of these datasets.
>
> ### 7. Strengthen the overall contributions by summarizing some takeaways from Table 11 and others.
>
> We have added an additional contribution regarding the detailed analysis conducted for ToMoE, including the results presented in Table 11 and other related findings.
>
> ### 8. Formats.
>
> We corrected the formatting issues pointed out by the reviewer.

---

> > ### Comment · Reviewer_o8BZ · 2025-11-10
> > **Thanks for the response and revision**
> >
> > The revision has addressed most of my concerns. However, I'm still curious about the MMLU performance comparison. Could you please provide the MMLU results under different activation ratio settings? For example, the zero-shot task performance as belows, or a simple line graph:
> >
> > |Methods|MMLU|ARC-e|ARC-c|PIQA|WinoG.|HellaS.|Average|
> > |:-|:-:|:-:|:-:|:-:|:-:|:-:|:-:|
> > |LLaMA-3 8B 100% (Dense)||
> > |LLaMA-3 8B 70% (ToMoE)||
> > |LLaMA-3 8B 60% (ToMoE)||
> > |LLaMA-3 8B 50% (ToMoE)||
> > |LLaMA-3 8B 40% (ToMoE)||

---

> > > ### Author Response · Authors · 2025-11-10
> > >
> > > Thank you very much for your response. Since the authors responsible for the rebuttal and revision are currently traveling and do not have access to the model or computational resources needed for training and evaluation, we will provide these results once we return later this week.

---

> > > > ### Comment · Reviewer_o8BZ · 2025-11-11
> > > >
> > > > Thanks for the reply. Yes, no problems. I think you could just evaluate LLaMA-3 8B ToMoE checkpoints in Table 3 with 70% and 60% activation ratios and update the results to Table 6.

---

> > > > > ### Author Response · Authors · 2025-11-14
> > > > >
> > > > > Thank you very much for your understanding. We have just finalized the MMLU results for the given active parameter settings. We added a new figure in the appendix (Figure 10) to illustrate the MMLU accuracy–parameter trade-off for the LLaMA-3 8B model. As you suggested, we also provide the corresponding numerical results below:
> > > > >
> > > > > | Methods                  | MMLU (5-shot, acc) | ARC-e (acc-norm) | ARC-c (acc-norm)| PIQA (acc-norm) | WinoG. (acc)| HellaS. (acc-norm) | Average |
> > > > > |--------------------------|------|-------|-------|------|--------|---------|---------|
> > > > > | LLaMA-3 8B 100% (Dense)  |  67.22     |    77.69   |    53.58   |   80.63   |    72.69    |     79.16    |  71.83       |
> > > > > | LLaMA-3 8B 70% (ToMoE)   |   49.46   |    71.55   |   44.71  |   76.28   |   68.98     |     71.84    |   63.14      |
> > > > > | LLaMA-3 8B 60% (ToMoE)   |   41.17   |   65.87    |   41.64    |   73.61   |    63.30    |     66.42    |  58.00       |
> > > > > | LLaMA-3 8B 50% (ToMoE)   |   36.31   |   60.56    |   38.04    |   71.27   |    58.95    |   61.61      |  54.12       |
> > > > > | LLaMA-3 8B 40% (ToMoE)   |   27.23   |   52.57    |    32.42   |    66.81  |    55.72    |   48.91    |  47.94       |

---

> > > > > > ### Comment · Reviewer_o8BZ · 2025-11-14
> > > > > >
> > > > > > Thanks a lot for the update! The results demonstrate that ToMoE degrades linearly with fewer activated parameters, which is promising. I believe the revision has now addressed all my concerns.

---

> > > > > > > ### Author Response · Authors · 2025-11-14
> > > > > > >
> > > > > > > Thank you again for your thoughtful feedback!

---

### Review · Reviewer_N19L · 2025-10-27

**Summary Of Contributions:**

### Summary
This paper proposes ToMoE, a novel method to convert dense models to MoE while mitigating performance loss and taking care of high computational cost. The core summary is as follows. ToMoE pioneers converting dense LLMs into sparse MoE models without updating the original model weights. It leverages dynamic structural pruning to "uncover" inherent expert structures within dense models. ToMoE solves a critical limitation of traditional dynamic pruning by ensuring fixed per-token active parameters.
### Strenths
The following are key strengths of this paper.
1. Low performance cost. ToMoE avoids the high fine-tuning cost of structural pruning (which requires expensive retraining to recover performance losses). ToMoE freezes original model weights and only trains lightweight components (routers, projection modules, hypernetwork), adding minimal parameters.
2. Practical deployment advantages, for exapmle, fixed latency. Unlike dynamic pruning (variable costs per input), ToMoE’s fixed per-token budget simplifies deployment in latency-sensitive applications.
3. Relatively high reproducibility. Provides detailed method introduction and specific implementation introduction.

### Weaknesses
However, this paper has some key shortcomings that would be greatly improved if the authors had the opportunity to further clarify or revise them. The following are key weaknessesof this paper.
1. The formatting of the paper requires significant adjustments. This may not be relevant to comprehension, but it will have a significant impact on rigor. Detailed information is provided in the following sections.
2. Expert count sensitivity. Table 3 show that increasing N beyond a threshold (e.g., N=16 for Qwen-2.5 7B) provides minimal or no performance gain, even introducing training burdens. This means users must tune N for each model, adding hyperparameter optimization costs.
3. The baselines and evaluations could be more comprehensive. Table 4 only covers two MoE construction baseline methods. A detailed investigation of more recent dense-to-MoE baselines, such as CMoE and DIVE, in subsequent versions would significantly enhance the paper's persuasiveness.
In addition, some in-depth experiments on the ToMoE mechanism can also add depth to the paper. For example, the authors can try to visualize and study whether ToMoE has a preference for specific parameters or heads during the pruning process, and try to derive more insights.

**Audience:**

Yes

**Audience Explanation:**

The paper’s core insights revolve around uncovering latent experts in dense LLMs via dynamic structural pruning to build efficient MoE models, without sacrificing performance or incurring heavy fine-tuning costs.
1. Dense LLMs inherently contain latent experts, no weight updates needed. Unlike prior MoE construction methods, ToMoE reveals that dense LLMs already have functional experts embedded in their layers. These experts can be extracted purely via dynamic structural pruning, avoiding the computational overhead of re-training model weights.
2. Dynamic pruning unifies expert construction and router training. Traditional MoE methods split expert building and router learning into two separate stages, leading to suboptimal coordination. ToMoE leverages dynamic pruning to merge these steps.
3. Fixed per-token computational budget solves dynamic pruning’s key flaw. Prior dynamic pruning methods lack a fixed budget for different inputs, making them incompatible with mini-batch inference or pre-filling (common in LLM deployment). ToMoE’s MoE conversion enforces a fixed per-token budget, balancing efficiency and deployment practicality.
4. No fine-tuning, yet SOTA performance across models. ToMoE outperforms sota structural pruning (e.g., LLM-Pruner, SliceGPT) and MoE construction methods (e.g., LLaMA-MoE, G-MoEfication) on perplexity and zero-shot tasks—without updating the original dense model’s weights.

**Broader Impact Concerns:**

Overall, this paper's basic reproducibility and methodological description are sufficient. The paper provides detailed implementation details, including hyperparameters and training methods. Beyond that, some minor issues may be considered:
1. A broader comparison of dense-to-MoE baseline methods would enhance the authority and dissemination of this paper's methods, as mentioned in the previous section.
2. A wider range of model sizes (larger dense models) would enhance the richness of the evaluation.
3. The paper does not discuss efficiency issues.

**Claims And Evidence:**

Yes

**Claims Explanation:**

The paper's core claims include four main areas: 1. ToMoE can convert dense LLMs into MoEs through dynamic structural pruning without updating the original model weights; 2. Joint optimization of the routing module and expert configuration enables efficient MoE construction; 3. This method outperforms state-of-the-art pruning and MoE methods on multiple models (Phi-2, LLaMA-2/3, Qwen-2.5) and multiple tasks without fine-tuning; 4. Three types of regularization, namely expert union, parameters, and load balancing, are crucial for performance.

Overall, Tables 3 and 4 demonstrate the effectiveness of the paper's approach and show that ToMoE can maintain good performance even with a lower activation ratio. However, as shown in the weaknesses in the previous section, a richer baseline and more thorough research would significantly improve the paper's quality.

**Requested Changes:**

Overall, this paper is of acceptable quality. It is suggested that the supplementary work mentioned in the previous sections can improve the quality of the paper. **However**, the following formatting issues must be brought to the attention of the authors. These are irrelevant to the understanding of the paper, but can significantly affect its rigor:
1. Table 5 has a formatting issue. I believe the author intended the left side to have a regular color display, but the current state makes the whole effect very imprecise.
2. Conventionally, mathematical formulas must end with punctuation. Some formulas in this paper have punctuation, but others do not. For example, Eq 5-10 is correct, but Eq 1-4 is not.
3. The figures formats in the paper are not uniform. Some figures were imported as pdfs, while others were imported as pngs. For example, the three images in Figure 5 are in different formats.

---

> ### Author Response · Authors · 2025-11-09
> **Response to Reviewer N19L**
>
> We want to thank Reviewer N19L for giving constructive feedback regarding our paper. We provided the following response below and **in the revised version of our paper**.
>
> ### 1. The formatting of the paper requires significant adjustments.
>
> Thank you very much for the detailed checking. We have (1) revised Table 5 (Table 7 in our revised paper) to ensure the formatting is correct, (2) ensured all equations have punctuation, and (3) replaced some figures so that all of them are in PDF now.
>
> ### 2. Expert count sensitivity.
>
> Thank you very much for pointing this out. We added one sentence to recommend choosing the number of experts to be smaller than 16 in our revised version.
>
> ### 3.  The baselines and evaluations could be more comprehensive.
>
> We further compare our method with CMoE and LLaMA-MoE-v2 in Tables 5 and 6 of our revised paper, following the evaluation protocols described in their original paper.
>
> ### 4. The paper does not discuss efficiency issues.
>
> We added the comparison of the throughput of ToMoE, LLaMA-MoE, and LLaMA-2 7B model in Table 15 of the appendix. ToMoE achieves throughput improvements, particularly when the mini-batch size is large.
>
>
> ### 5. A wider range of model sizes.
>
> Thank you for the suggestion. We have included the 13B and 14B variants of LLaMA-2 and Qwen-2.5 in our original submission. However, due to resource constraints, it is difficult for us to obtain sufficient GPU capacity to further scale our experiments to larger model sizes.

---

### Review · Reviewer_65R1 · 2025-10-29

**Summary Of Contributions:**

1. The work proposes ToMoE, which introduces **a novel structured pruning approach** by converting dense LLMs into MoE models via dynamic structural pruning, achieving consistently strong performance without weight updates.
2. ToMoE offers a **low-cost alternative to build MoE from dense LLMs**, avoiding expensive fine-tuning or pretraining, compared with Llama-MoE[1][2].
3. ToMoE features a **unified, efficient training scheme** with joint expert and router optimization, plus tailored strategies for MLP (top-1) and MHA (top-K + static) layers.
4. Extensive experiments prove its outperforms state-of-the-art pruning methods across multiple models (LLaMA-2/3, Qwen-2.5) in both perplexity and zero-shot tasks.


- [1] LLaMA-MoE: Building Mixture-of-Experts from LLaMA with Continual Pre-training.
- [2] LLaMA-MoE v2: Exploring Sparsity of LLaMA from Perspective of Mixture-of-Experts with Post-Training

**Audience:**

Yes

**Audience Explanation:**

This work proposes a new method in the field of structure pruning, and there is still a lot to explore. It opens up a promising direction for future research in model efficiency and sparsity, and we expect it to inspire a wave of follow-up studies exploring dynamic, parameter-efficient architectures without full fine-tuning.

**Broader Impact Concerns:**

1. This work presents a technical method for converting dense large language models into Mixture-of-Experts architectures for dynamic structural pruning. It does not introduce new capabilities (e.g., generation of harmful content, surveillance, or decision-making in sensitive domains) beyond those already present in the original models.
2. This work does not involve human subjects, personal data, or deployment in high-stakes applications; we believe it raises no significant ethical or societal concerns that warrant a detailed Broader Impact Statement.
3. The primary impact is positive: enabling more efficient and accessible deployment of existing LLMs with reduced computational cost.

**Claims And Evidence:**

Yes

**Claims Explanation:**

1. **ToMoE outperforms existing structural pruning methods while maintaining or reducing computational cost**: This is strongly supported by empirical evaluations across multiple model families (LLaMA-2, LLaMA-3, Qwen-2.5) and tasks. Tables 1 and 3 present perplexity and zero-shot performance metrics showing that ToMoE consistently achieves lower perplexity and higher accuracy than state-of-the-art baselines like LLM-Pruner, SliceGPT, LLM Surgeon, and DISP-LLM, even with fewer active parameters.

2. **ToMoE effectively converts dense models into MoE models without fine-tuning the original weights**: The ablation study in Figure 5(c) shows that removing this self-knowledge distillation component significantly degrades performance, confirming its importance in preserving model capacity without updating the core weights. The union of experts regularization (R_U) is shown in Figure 5(a) to ensure that collectively, the experts utilize nearly all original parameters, addressing concerns about capacity loss from pruning. Table 5(b) proves its superior efficiency in existing structure purning methods.

3. But the claims of the advantages of ToMoE as MoE construction method still lack sufficient proof, which will be explained in 'Request Changes'.

In sum, the claims in this paper are mostly supported with visualizations or experiments, and are very convincing.

**Requested Changes:**

1. ToMoE currently only uses top-1 routing for MLP layers in ToMoE, but exploring top-k routing is both possible and might be helpful.
2. Table 4 is supposed to show that ToMoE is the better MoE construction method, but the comparison isn't fair. Llama-MoE is an example of this. Llama-MoE is randomly initialised and needs extensive continue-pretraining, but in this work, it is evaluated before training. The '+finetune' result is only trained with equal iteration with ToMoE, which is far less than the required training cost of Llama-MoE. It is wrong to say that ToMoE is better than LLaMA-MoE when the number of training steps is the same, because LLaMA-MoE was not designed to work with few training steps. The author should try to show that ToMoE can achieve performance close to LLaMA-MoE, even with a simpler training process and a lower training budget. It is very important to explicitly evaluate this comparison. As an alternative, you can compare your results directly with the officially released LLaMA-MoE checkpoint.
3. Llama-MoE v2[2] is another advanced MoE construction method. I would like to see a comparison.

Overall, I believe that ToMoE is an promising work in terms of structure pruning, but its advantage as a MoE construction method still lacks sufficient proof.

- [1] LLaMA-MoE: Building Mixture-of-Experts from LLaMA with Continual Pre-training.
- [2] LLaMA-MoE v2: Exploring Sparsity of LLaMA from Perspective of Mixture-of-Experts with Post-Training

---

> ### Author Response · Authors · 2025-11-09
> **Response to Reviewer 65R1**
>
> We want to thank Reviewer 65R1 for giving constructive feedback regarding our paper. We provided the following response below and **in the revised version of our paper**.
>
> ### 1. Expand to top-k routing.
>
> We agree with Reviewer 65R1 that extending our approach to top-$k$ routing in MLP layers is a promising future direction. However, we would like to emphasize that such an extension is not as straightforward as it may appear. There are two main challenges: (1) designing an effective and efficient differentiable top-$k$ operator. Although several differentiable top-$k$ variants exist, they are generally less stable and efficient compared to the Gumbel-Softmax used in our current formulation, and (2) managing overlap among the selected $k$ experts. When multiple experts are activated simultaneously, overlapping regions between their subspaces appear, which can no longer be formulated as a dynamic structural pruning problem. We have conducted preliminary explorations toward a top-$k$ extension, but do not yet have concrete results. We believe that fully developing this direction requires a dedicated follow-up project.
>
> ### 2. &3. Compare with more extensively trained LLaMA-MoE and comparison with LLaMA-MoE-v2.
>
> We have added comparisons with LLaMA-MoE trained on 1.2B tokens (Table 5 in our revision) and with the fully trained LLaMA-MoE-v2 trained on 7B tokens (Table 6 in our revision). In these new results, ToMoE continues to perform on par with or better than both LLaMA-MoE and LLaMA-MoE-v2. However, we believe our routing-only method and a fully retrained LLaMA-MoE model are not directly comparable, as closing the performance gap resulting from 200B-token fine-tuning is practically impossible without updating the model weights.

---

### Decision · Action_Editor_8aJW · 2025-12-14

**Recommendation:** Accept as is

**Additional Comments:**

The authors have made revisions during the rebuttal period by incorporating reviewers' comments. Please submit the final version.

**Audience:**

Yes

**Audience Explanation:**

The paper proposed a well-motivated, efficient dynamic pruning strategy for MoE model building. The work could be beneficial to the reseachers that are in LLMs, model architectures.

**Claims And Evidence:**

Yes

**Claims Explanation:**

The paper has made claims about the novelty and the contribution of the proposed method. The claim that the efficient dynamic pruning strategy is effective has been supported by comprehensive empirical studies, fair comparisons with other baselines, and deep analysis.